🔓 | **Open Peer Review** | Parasitology | Research Article

# Compound Biejia-Ruangan tablets activate the STING-TBK1 pathway to alleviate hepatic fibrosis in alveolar echinococcosis

Yuyu Ma,[1] Hui Zhao,[2] Yumei Liu,[3] Menggen Meng,[1] Jiahui Chen,[1] Yuqin Sun,[1] Bin Fan,[4] Xuan Zhou,[1] Fengmin Tian,[1] Xiumin Ma,[1] Liang Wang[1,5]

**ABSTRACT** Alveolar echinococcosis (AE) is a lethal parasitic disease that leads to progressive liver fibrosis, a process characterized by the dominance of M2 macrophages in advanced stages. While the compound Biejia-Ruangan tablet (CBRT) has demonstrated efficacy in reversing hepatic fibrosis, its underlying mechanisms in the context of AE-induced fibrosis are not fully understood. This study aimed to investigate the therapeutic potential of CBRT against AE-induced liver fibrosis and to elucidate its mechanisms of action. Serum levels of liver fibrosis markers and cytokines were measured by ELISA. Liver histopathology and collagen deposition were assessed using H&E and Masson's trichrome staining. Apoptosis in GHA1 hepatocytes was evaluated by flow cytometry. Macrophage migration was analyzed via scratch and Transwell invasion assays. The involvement of the STING-TBK1 signaling pathway was examined through western blotting (WB), immunohistochemistry (IHC), immunofluorescence (IF), and quantitative real-time PCR (qRT-PCR). Livers from *Echinococcus multilocularis* (*E.m*.)-infected mice exhibited severe fibrosis accompanied by significant infiltration of M2 macrophages and upregulated expression of STING-TBK1 pathway components (Arg-1, α-SMA, STING, and TBK1). Treatment with CBRT (1 g/kg) markedly attenuated liver fibrosis *in vivo* ($P < 0.01$) and inhibited the activation of mouse hepatic stellate cells (mHSCs) *in vitro*. CBRT suppressed the activation of HSCs, partly by upregulating IFN-γ, suppressed apoptosis in GHA1 hepatocytes, and improved hepatocyte function. Mechanistically, CBRT activates the STING-TBK1 signaling pathway, which subsequently suppresses the polarization of macrophages toward the M2 phenotype, thereby alleviating fibrogenesis. This study demonstrates that CBRT alleviates hepatic fibrosis by activating the STING-TBK1 signaling pathway. This activation elicits a shift in macrophage polarization from the pro-fibrotic M2 phenotype towards the M1 phenotype, which subsequently suppresses the activation of hepatic stellate cells and alleviates fibrogenesis. These findings reveal a novel immunomodulatory mechanism of CBRT centered on the STING-TBK1 axis, providing both experimental evidence and a theoretical foundation for its potential application in treating hepatic fibrosis associated with AE.

**IMPORTANCE** This study demonstrates that compound Biejia-Ruangan tablet (CBRT) alleviates *Echinococcus multilocularis* (*E.m*.)-induced hepatic fibrosis by reactivating the suppressed STING-TBK1 signaling pathway in the chronic infection stage, thereby reprogramming macrophage polarization from a pro-fibrotic M2 phenotype toward an M1 phenotype. This work identifies CBRT as a promising immunomodulatory agent that can reverse the Th2/M2-dominated immunosuppressive microenvironment, providing a novel therapeutic strategy for AE and potentially other infection-associated fibrotic diseases. However, further research is required to validate the translational efficacy of CBRT and to define the precise therapeutic window for STING-TBK1 modulation in human fibrotic disorders.

**Peer Reviewer** Claudio Vieira da Silva, Universidade Federal de Uberlandia, Uberlandia, Brazil

Address correspondence to Xiumin Ma, maxiumin1210@sohu.com, or Liang Wang, 16206441@qq.com.

Yuyu Ma, Hui Zhao, Yumei Liu, and Menggen Meng contributed equally to this article. Author order was determined by mutual agreement based on their respective contributions.

The authors declare no conflict of interest.

**KEYWORDS** compound Biejia-Ruangan tablet, hepatic fibrosis, STING-TBK1 signaling pathway

Alveolar echinococcosis (AE), a lethal zoonotic parasitic infection, presents a significant public health burden, particularly in endemic regions such as western China (1). The disease, caused by the larval stage of *Echinococcus multilocularis* (*E.m.*), is characterized by tumor-like, infiltrative growth primarily in the liver. This invasive progression, often termed "parasitic cancer," leads to extensive tissue destruction and metastasis, resulting in a 10-year mortality rate exceeding 90% if untreated (1, 2). Surgical resection offers the only potential cure, but its efficacy is limited by late diagnosis—due to an often-asymptomatic early phase—and high post-operative recurrence rates (3, 4). A key driver of this relentless progression is the establishment of a pro-fibrotic and immunosuppressive microenvironment within the liver. Notably, the polarization of infiltrating macrophages toward an M2 phenotype supports parasitic persistence and actively promotes the deposition of extracellular matrix, exacerbating liver fibrosis (5, 6). This fibrotic response not only facilitates parasite survival but also contributes to organ failure, highlighting an urgent need for novel therapeutic strategies that can modulate the host immune response and halt fibrogenesis in AE.

In the search for antifibrotic therapies, traditional Chinese medicine has yielded promising candidates. Among them, the compound Biejia-Ruangan tablet (CBRT) is notable as the first herbal formulation approved by the China Food and Drug Administration specifically for the treatment of hepatic fibrosis (7). Clinical and preclinical studies have established that CBRT can inhibit hepatic stellate cell (HSC) activation, reduce collagen I synthesis and deposition, and promote the resolution of existing fibrotic scars, thereby attenuating and even reversing early-stage liver cirrhosis (8, 9). Given this broad-spectrum antifibrotic efficacy, we hypothesized that CBRT could also ameliorate liver fibrosis in the context of AE. However, its precise mechanism of action in this parasitic infection, particularly concerning immune modulation within the unique AE microenvironment, remains completely unexplored.

Liver fibrosis is a dynamic process regulated by a complex interplay of immune cells, with macrophages serving as central orchestrators (10). Their functional plasticity allows them to adopt either pro-fibrotic (M2-like) or anti-fibrotic (M1-like) phenotypes, critically influencing disease outcomes. Recent evidence highlights the stimulator of interferon genes (STING) pathway as a pivotal regulator of innate immunity and macrophage polarization. Cytosolic DNA sensing via the cGAS enzyme leads to the production of the second messenger 2′,3′-cGAMP, which activates STING. This triggers a signaling cascade involving TANK-binding kinase 1 (TBK1) and the transcription factor IRF3, ultimately driving the production of type I interferons (IFNs), such as IFN-β (11, 12). IFN-β is a potent inducer of classical M1 macrophage activation, promoting a robust antimicrobial and inflammatory state (13). Intriguingly, while the cGAS-STING pathway is a well-defined defense mechanism against viral and intracellular bacterial infections, its role—and potential therapeutic manipulation—in parasitic diseases like AE is poorly understood.

In this study, we demonstrate that CBRT exerts a potent therapeutic effect against AE-induced liver fibrosis. Treatment with CBRT attenuates hepatic fibrosis, suppresses the activation of HSC, and confers hepatoprotection. Mechanistically, our findings reveal that CBRT alleviates liver fibrosis by activating the STING-TBK1 signaling pathway, which in turn inhibits M2 macrophage polarization.

## MATERIALS AND METHODS

### Human subjects

Human liver tissues were collected from 21 patients diagnosed with AE who underwent surgical resection. All patients provided written informed consent, and the AE diagnosis was histologically confirmed for each case. For analysis, tissue specimens were categorized into two groups: close lesion tissue (CLT) samples, collected from within 2 cm

of the lesion border, and distant lesion tissue (DLT) samples, collected from at least 2 cm outside the lesion margin. The study protocol received ethical approval from the Institutional Review Board of the First Affiliated Hospital of Xinjiang Medical University (approval no. 20190225-29, Urumqi, China).

## Animals

Female BALB/c mice (6–8 weeks old) of specific pathogen-free grade were obtained from the Animal Experiment Center of the First Affiliated Hospital of Xinjiang Medical University. All animal experiments were performed in strict compliance with institutional guidelines and were approved by the Animal Care and Use Committee of the First Affiliated Hospital of Xinjiang Medical University (approval no. IACUC-20230307-16, Urumqi, China).

## Effect of drug-containing plasma on the proliferation of mHSC

The intervention doses and frequencies for mice in groups T1 (high dose), T2 (medium dose), and T3 (low dose) were determined based on the clinical dosage of CBRT using standard body surface area conversion factors. Twenty BALB/c mice were randomly divided into four groups ($n = 5$ per group): a saline control group and three CBRT treatment groups. The treatment groups received daily doses via oral gavage on different schedules: the high-dose group (T1, 2 g/kg) at 9:00; the medium-dose group (T2, 1 g/kg) at 9:00 and 13:00; and the low-dose group (T3, 0.5 g/kg) at 9:00, 13:00, and 17:00. Serum was collected from the retro-orbital plexus at 1, 2, 4, 6, 8, 10, 12, and 24 h after the initial dose, then processed (centrifugation, heat inactivation, and filtration) and stored at $-20°C$ for mHSC assays. mHSCs were plated in 96-well plates ($1 \times 10^4$ cells/well). After attachment and 48-h serum starvation, cells were exposed for 48 h to medium supplemented with 20% plasma from either CBRT-treated (T1, T2, and T3) or control mice. Cell viability was determined by MTT assay: following a 4-h incubation with MTT reagent, formazan crystals were dissolved in DMSO, and absorbance was read at 490 nm. The percentage inhibition of proliferation was calculated using the formula: $100 \times [1 - (A0/A1)]$, where A-sample and A-control are the mean absorbance values of treated and control wells, respectively.

$$\text{Inhibition rate (E)} = \frac{A0 - A1}{A0} \times 100\%$$

## E. m. infection mouse model and treatment specimen collection

Protoscoleces of *E.m.* (PSCs) were aseptically isolated from gerbil-maintained stocks and used to establish a quantitative *E.m.* infection model by intrahepatic inoculation of 2,000 PSCs per mouse. Control mice received an equivalent volume of sterile saline via intraperitoneal injection. To dynamically monitor infection progression, mice in the infection model group were euthanized at 8, 30, 60, and 90 days post-infection ($n = 5$ per time point). To evaluate the therapeutic effect of CBRT, a separate treatment group of infected mice received daily oral gavage of CBRT from day 60 to day 90 post-infection. The dosing regimens (low Biejia dose: 0.5 g/kg; high Biejia dose: 1 g/kg) were determined based on the clinical human dose using standard body surface area conversion factors.

## ELISA detection

Serum levels of aspartate aminotransferase (AST), alanine aminotransferase (ALT), four specific liver fibrosis markers, Four Liver Fibrosis Markers (hyaluronic acid [HA], laminin [LN], procollagen-III-peptide [PCIII], and collagen-type-IV [C-IV]) and cytokines were quantified in samples collected from the retro-orbital plexus of mice in the *E.m.* infection and CBRT treatment groups. Measurements were performed using ELISA kits (Saipai Biology, China) according to the manufacturer's protocols. Detailed information on the antibodies used in the ELISA in Table S1.

## *E.m*. antigen preparation (EmP)

EmP was prepared by cryogenic grinding of PSCs in liquid nitrogen until a homogeneous powder was obtained. The powder was suspended in a suitable buffer and homogenized by sonication using 5-s pulses for a total duration of 2 min. The resulting homogenate was then rotated end-over-end at 4°C overnight to facilitate antigen extraction. Finally, the homogenate was centrifuged at $12,000 \times g$ for 30 min at 4°C to clarify the supernatant, which was collected as the soluble EmP extract.

## Cell culture

The mouse hepatocyte line GHA1 and mHSCs (BeNa Culture Collection) were routinely maintained in RPMI-1640 medium. The RAW264.7 macrophage line (Pricella) was maintained in high-glucose DMEM. Both media were supplemented with 10% fetal bovine serum (FBS), 100 U/mL penicillin, and 100 µg/mL streptomycin. For co-culture experiments, cells were washed and resuspended in high-glucose DMEM. GHA1 cells ($1 \times 10^6$ cells/mL) were seeded in the upper chamber of Transwell inserts, while RAW264.7 cells ($1 \times 10^6$ cells/mL) were seeded in the lower chamber. Co-cultures were assigned to the following treatment groups for 48 h: (i) NC group: PBS vehicle; (ii) Em group: stimulation with 100 µg/mL EmP; (iii) Em + High Biejia: EmP stimulation plus serum from mice treated with CBRT at 1 g/kg; and (iv) Em + Low Biejia: EmP stimulation plus serum from mice treated with CBRT at 0.5 g/kg.

## Cell invasion assay

A Transwell invasion assay was performed using chambers with 8 µm pore membranes. Briefly, RAW264.7 cells ($5 \times 10^5$ cells/well) in serum-free medium were seeded into the upper chamber. The lower chamber was filled with complete medium (containing 10% FBS) to establish a chemotactic gradient. After 24 h of incubation at 37°C, cells on the upper membrane surface were gently removed. Cells that had migrated to the lower surface were fixed, stained with 0.1% crystal violet, and imaged. The number of invaded cells was quantified by counting cells in five random fields per well under a light microscope.

## Cell scratch assay procedure

RAW264.7 cells were seeded in pre-marked six-well plates at $5 \times 10^5$ cells/well and incubated until 90% confluent. A uniform scratch was created across each well using a sterile 200 µL pipette tip, guided by reference lines on the plate underside. After washing with PBS to remove debris, serum-free medium was added. Scratch images were captured at 0 and 24 h post-scratch using phase-contrast microscopy to quantify cell migration.

## Annexin V-FITC apoptosis detection assay

GHA1 hepatocytes, treated for 24 h with serum from CBRT-treated mice, were stained using the FITC Annexin V/PI Apoptosis Detection Kit (BD Biosciences, 556547) according to the manufacturer's protocol. Briefly, $1 \times 10^5$ cells were stained with Annexin V-FITC and PI in binding buffer for 15 min in the dark. Flow cytometry was performed within 1 h. Cells were quantified as viable (Annexin V−/PI−), early apoptotic (Annexin V+/PI−), or late apoptotic/necrotic (Annexin V+/PI+).

## Histopathological analysis

Liver tissues from AE patients and infected mice were fixed, dehydrated through a graded ethanol series, and embedded in paraffin. Subsequently, 5-µm-thick sections were cut and stained with hematoxylin and eosin (H&E), Masson's trichrome (using a commercial kit; Shanghai Maokang, MM1007), and picrosirius red (using a commercial

kit; Shanghai Maokang, MM1036) according to standard protocols to assess general histology and collagen deposition.

## Immunohistochemistry (IHC) and immunofluorescence (IF) analysis

Paraffin sections underwent standard antigen retrieval and blocking. For IHC, sections were incubated with primary antibodies overnight at 4℃, followed by HRP-conjugated secondary antibodies and DAB visualization. For IF, sections were incubated with primary antibodies and then with fluorescent secondary antibodies (goat anti-mouse Alexa Fluor 488, 1:200; goat anti-rabbit Alexa Fluor 647, 1:200) for 2 h at room temperature in the dark. For quantitative analysis, three non-overlapping fields of view were randomly captured from both lesion areas and adjacent parenchyma for each IHC/IF-stained section under consistent microscopic parameters. Following uniform background correction and spatial calibration, all images were anonymized and recoded to enable blinded analysis. Quantitative assessment was performed using ImageJ software: for IHC images, a uniform color threshold was applied to segment positive staining areas, while for IF images, a fluorescence intensity threshold was set to identify specific signals. The positive area percentage (Area%) was calculated for each field of view. The final quantitative value for each independent biological sample (*n* corresponds to an individual mouse; specific *n* values are provided in the figure legends) was determined as the mean Area% from its three fields. Detailed information on the specific antibodies used in this study, including the target antigen, vendor/supplier, catalog number, and dilution, is provided in Table S1.

## RNA isolation and real-time PCR (qRT-PCR)

Total RNA was extracted from human AE tissues, mouse liver tissues, and *in vitro* cell cultures using TRIzol reagent. RNA was reverse transcribed into cDNA using the PrimeScript RT Reagent Kit (Takara, RR047A). Quantitative real-time PCR was performed using TB Green Premix Ex Taq II (Takara, RR820A) on a 7500 Fast Real-Time PCR System (Applied Biosystems). Gene expression levels were normalized to GAPDH, and primer sequences are listed in Table S1.

## Western blot analysis

Total protein was extracted from tissues and cells using RIPA lysis buffer supplemented with protease and phosphatase inhibitors. Protein concentration was determined using a bicinchoninic acid assay. Equal amounts of protein were separated by SDS-PAGE on 10–12% gels and transferred to polyvinylidene difluoride membranes. Membranes were blocked with 5% non-fat milk and incubated overnight at 4℃ with primary antibodies (Table S1), followed by incubation with appropriate HRP-conjugated secondary antibodies at room temperature for 2 h. Protein bands were visualized using an enhanced chemiluminescence substrate and imaged with a ChemiDoc system. Band intensities were quantified using Image Lab software.

## Statistical analysis

Data are presented as the mean ± SD. Statistical analyses were performed using SPSS 29.0 and GraphPad Prism 9.0 software. Comparisons between two groups were made using a paired, two-tailed Student's *t*-test. For comparisons among three or more groups, one-way analysis of variance (ANOVA) was applied, followed by Tukey's post hoc test for multiple comparisons. A $P$ value of less than 0.05 was considered statistically significant ($*P < 0.05$, $**P < 0.01$, and $***P < 0.001$).

## RESULTS

### The liver lesion tissues of patients with AE have severe fibrosis

Patient baseline characteristics are summarized in Table 1. Histopathological analysis demonstrated that AE liver lesions are characterized by cyst-mediated destruction of the hepatic architecture, surrounded by an irregular, hyperplastic fibrous capsule (Fig. 1A and B). Masson's trichrome staining further confirmed extensive fibrosis, visualized as dense blue/green collagen bundles enveloping the lesions (Fig. 1A and C). IHC analysis confirmed that in AE patients, the CLT exhibited significantly greater positive staining areas for macrophage markers (iNOS for M1 and Arg-1 for M2) and key fibrogenic markers (α-SMA and Collagen I), as well as for components of the STING-TBK1 pathway, compared to DLT (Fig. 1D and E). These findings were further corroborated at the molecular level: WB and qRT-PCR analyses demonstrated significantly elevated protein and mRNA expression of iNOS, Arg-1, α-SMA, and key STING-TBK1 pathway molecules in CLT relative to DLT (Fig. 1F through H). Collectively, these data indicate that the perilesional microenvironment is characterized by robust macrophage infiltration, active fibrogenesis, and activation of the STING-TBK1 signaling axis.

### CBRT has a significant repairing effect on the liver function of mice with *E.m*.

As outlined in the experimental design (Fig. 2A), CBRT treatment significantly ameliorated liver injury in *E.m*.-infected mice, as evidenced by the dose-dependent reduction in serum levels of AST and ALT, with the high-dose group (1 g/kg) showing the most pronounced effect (Fig. 2B and C). Concurrently, CBRT administration markedly decreased the serum concentrations of key hepatic fibrosis markers—HA, LN, PCIII, and C-IV also in a dose-dependent manner (Fig. 2D through G). At the immunological level, CBRT altered the circulating cytokine profile, increasing the levels of IFN-γ and IL-2 (Fig. 2H and I) while decreasing those of IL-4 and IL-10 (Fig. 2J and K).

**TABLE 1** Baseline characteristics and laboratory profiles of study participants

| Characteristic | HAE (*n* = 21) |
|---|---|
| Gender | |
| Male | 12 (57.14%) |
| Female | 9 (42.86%) |
| Age, years | |
| ≤18 | 2 (9.52%) |
| 19–59 | 16 (76.19%) |
| ≥60 | 3 (14.29%) |
| BMI, kg/m$^2$ | |
| <25 | 16 (76.19%) |
| ≥25 | 5 (23.81%) |
| HBV | |
| Yes | 2 (9.52%) |
| No | 19 (90.48%) |
| Child-Pugh grade | |
| A | 10 (47.62%) |
| B | 10 (47.62%) |
| C | 1 (4.76%) |
| Laboratory tests ($X \pm s$) | |
| Cr, μmol/L | 58.64 ± 6.27 |
| Albumin, g/L | 37.56 ± 3.39 |
| Laboratory tests [Median (IQR)] | |
| TB, μmol/L | 12.80 (9.48, 19.79) |
| ALT, U/L | 26.30 (19.30, 67.20) |
| AST, U/L | 32.07 (18.08, 70.90) |

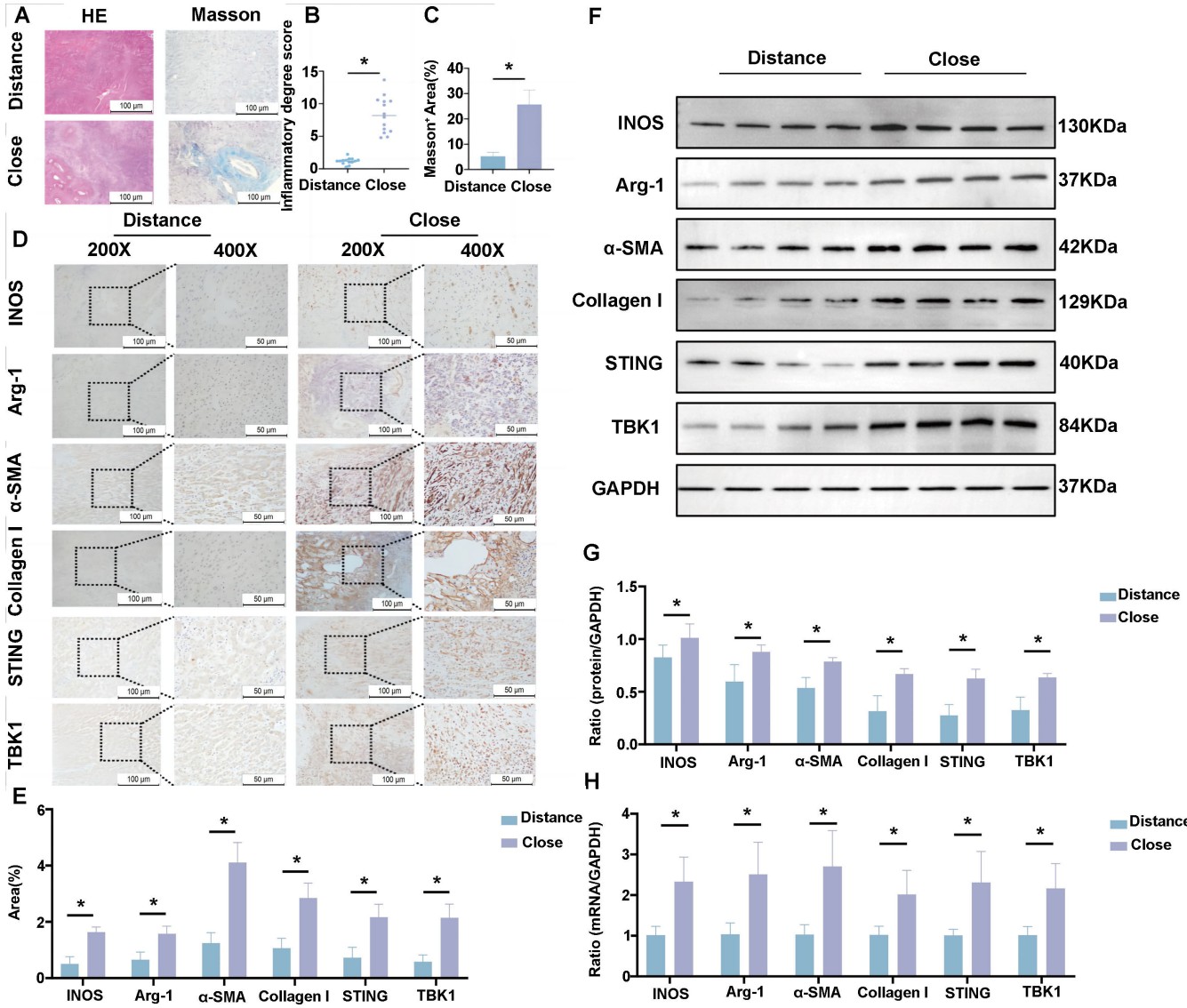

**FIG 1** Severe fibrosis in the liver lesion tissues of AE patients. (A) Representative images of H&E and Masson's trichrome staining of the CLT and DLT from AE patients. (B and C) Quantitative analysis of H&E and Masson's trichrome staining in CLT and DLT regions. (D and E) Representative IHC staining and corresponding quantification for iNOS, Arg-1, α-SMA, Collagen I, STING, and TBK1 in CLT and DLT regions. (F and G) Representative WB images and densitometric analysis of protein levels for iNOS, Arg-1, α-SMA, Collagen I, STING, TBK1, and GAPDH (loading control) in CLT and DLT regions. (H) mRNA expression levels of iNOS, Arg-1, α-SMA, Collagen I, STING, TBK1, and GAPDH (reference gene) in CLT and DLT regions, as determined by qRT-PCR. Data are from three biologically independent replicates ($n = 3$) and presented as mean ± SEM. Statistical significance was determined using unpaired Student's $t$-test (for comparisons between two groups) or one-way ANOVA (for multi-group comparisons followed by appropriate post hoc tests), as indicated: *$P < 0.05$.

## CBRT alleviates liver fibrosis in mice infected with *E.m*.

In mice with established 60-day *E.m*. infections, a subsequent 30-day course of CBRT significantly reduced both the number and volume of hepatic cysts compared to untreated controls (Fig. 3A). Histopathological analysis by H&E staining showed that CBRT treatment effectively attenuated the infection-induced pathology, including perilesional inflammatory cell infiltration, hepatocyte necrosis, and disruption of the lobular architecture, leading to a more organized hepatocellular arrangement (Fig. 3B and C). Masson's trichrome staining revealed that CBRT treatment attenuated hepatic fibrosis, as shown by reduced collagen deposition, shortened fibrous septa, decreased pseudo lobule formation, and a restored fibrotic-to-parenchymal ratio resembling earlier

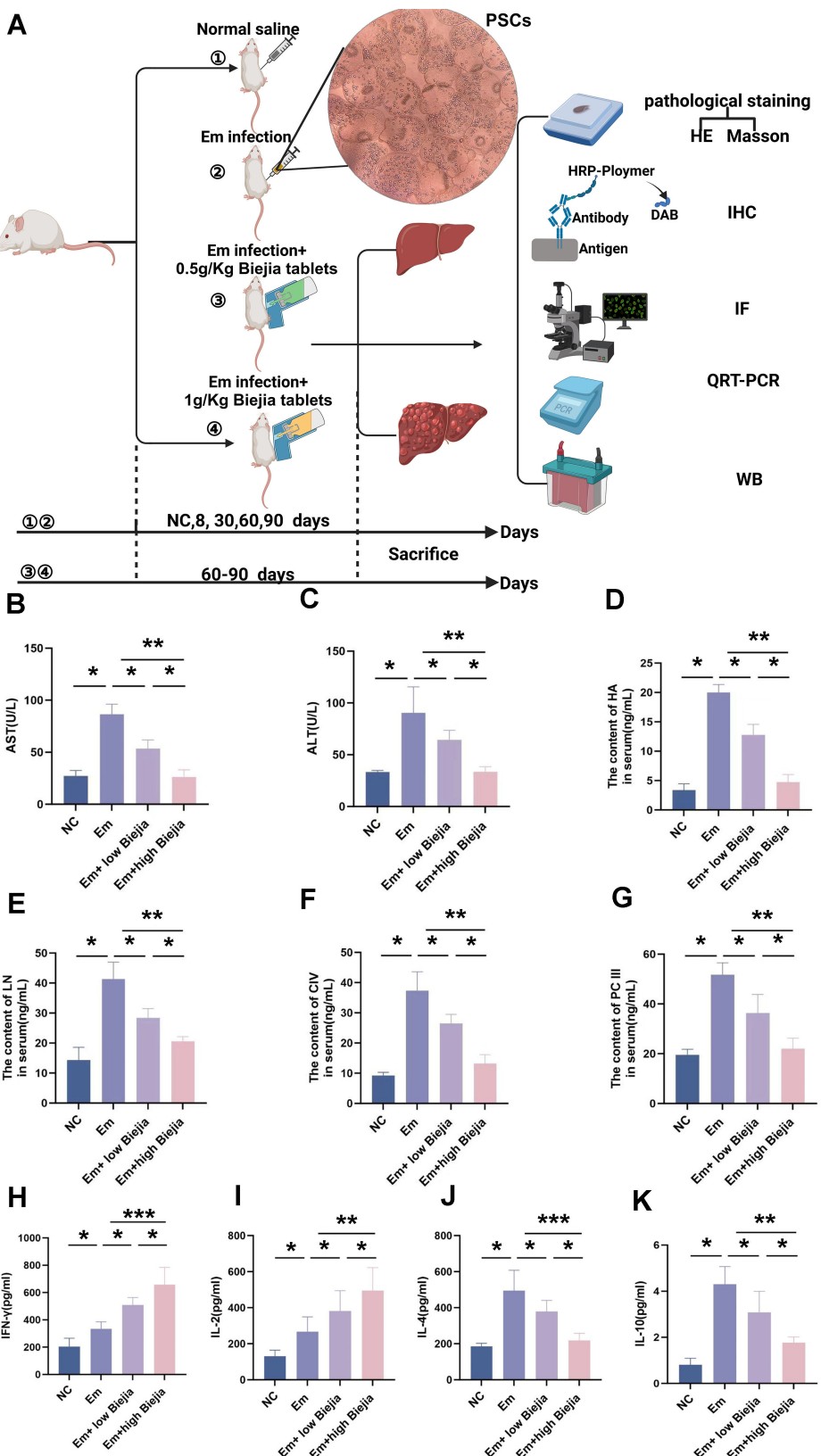

**FIG 2** CBRT ameliorates liver injury and alters serum fibrosis/immune markers in a murine model of AE. (A) Schematic diagram of the alveolar echinococcosis mouse model establishment and experimental grouping. (B and C) Serum levels of ALT and AST in each group, as measured by ELISA, with corresponding quantitative analysis. (D–G) Serum levels of

Fig 2 (Continued)

fibrosis markers, including PCIII, C-IV, LN, and HA, in each group were determined by ELISA and corresponding quantitative analysis. (H–K) Serum levels of cytokines, including IFN-γ, IL-2, IL-4, and IL-10, in each group were determined by ELISA and corresponding quantitative analysis. Data are from three biologically independent replicates ($n = 5$) and presented as mean ± SEM. Statistical significance was determined using unpaired Student's $t$-test (for comparisons between two groups) or one-way ANOVA (for multi-group comparisons followed by appropriate post hoc tests), as indicated: *$P < 0.05$, **$P < 0.01$, and ***$P < 0.001$.

infection stages. These morphological improvements indicate that CBRT inhibits the progression of $E.m.$-induced liver fibrosis (Fig. 3B and C). Furthermore, compared to the Em group, high-dose CBRT treatment reduced both the IHC staining area and the protein/mRNA expression levels of the pro-fibrotic markers Arg-1 and Collagen I (Fig. 3I through L, 4A, F and G L and M), consistent with an attenuation of hepatic fibrosis and structural improvement. In parallel, high-dose CBRT increased the IHC signal and the protein/mRNA expression of STING and TBK1 (Fig. 3E through H, 4A, D and E J through K, ), indicating activation of the STING-TBK1 signaling pathway.

## CBRT alleviates hepatic fibrosis in $E.m.$-infected mice by activating the STING-TBK1 signaling pathway and suppressing M2 macrophage polarization

In mice with established 60-day $E.m.$ infections, a subsequent 30-day treatment with CBRT (administered during the final 30 days of a total 90-day infection period) altered macrophage polarization compared to untreated mice at the 90-day endpoint. CBRT treatment decreased the expression of the M2 marker Arg-1 (Fig. 4A, C and I) and

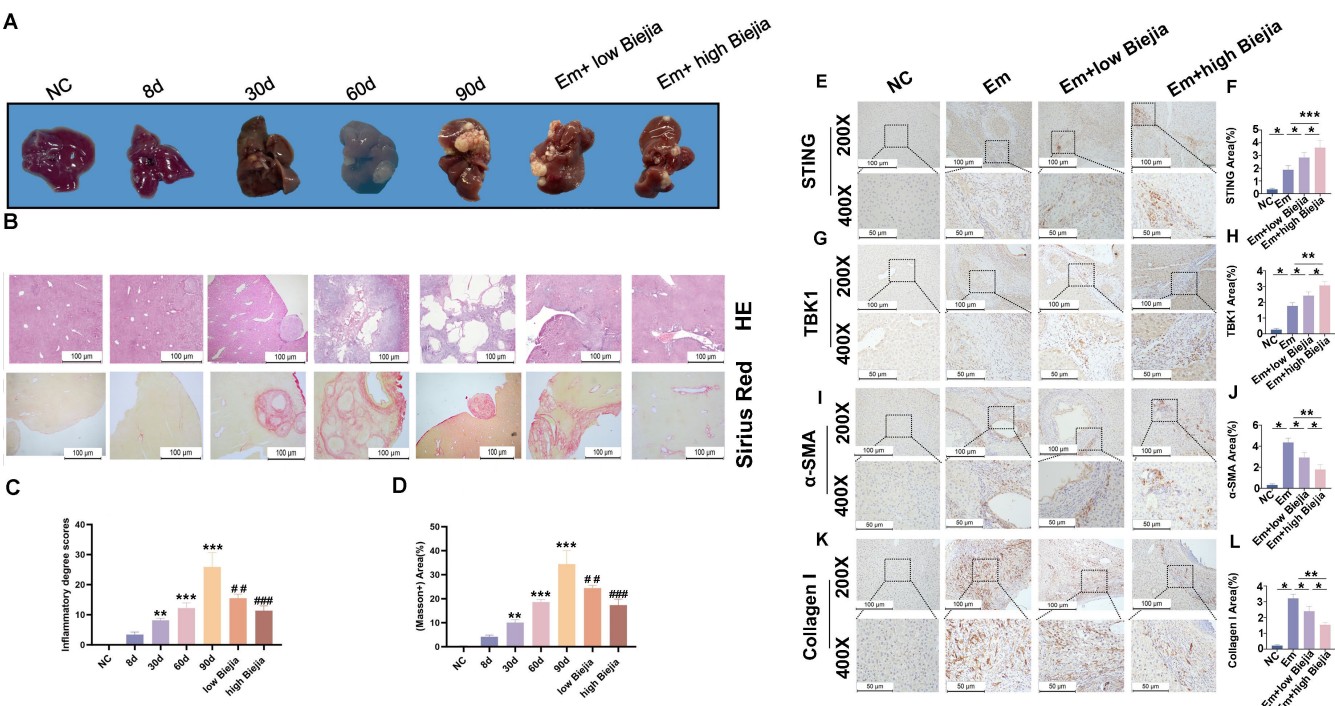

**FIG 3** CBRT mitigates hepatic histopathological changes and fibrosis-related protein expression in AE. (A) Gross morphology of livers from mice in each experimental group. (B–D) Representative photomicrographs of H&E and Sirius red staining of liver sections, with corresponding quantitative analysis of pathological injury and collagen deposition. (E–L) Representative IHC staining images for STING, TBK1, α-SMA, and Collagen I in liver tissues, with corresponding quantitative analysis of each marker. Data are from three biologically independent replicates ($n = 5$) and presented as mean ± SEM. Statistical significance was determined using unpaired Student's $t$-test (for comparisons between two groups) or one-way ANOVA (for multi-group comparisons followed by appropriate post hoc tests), as indicated: **$P < 0.01$ and ***$P < 0.001$ versus the control group; ##$P < 0.01$ and ###$P < 0.001$ versus the 90-day group.

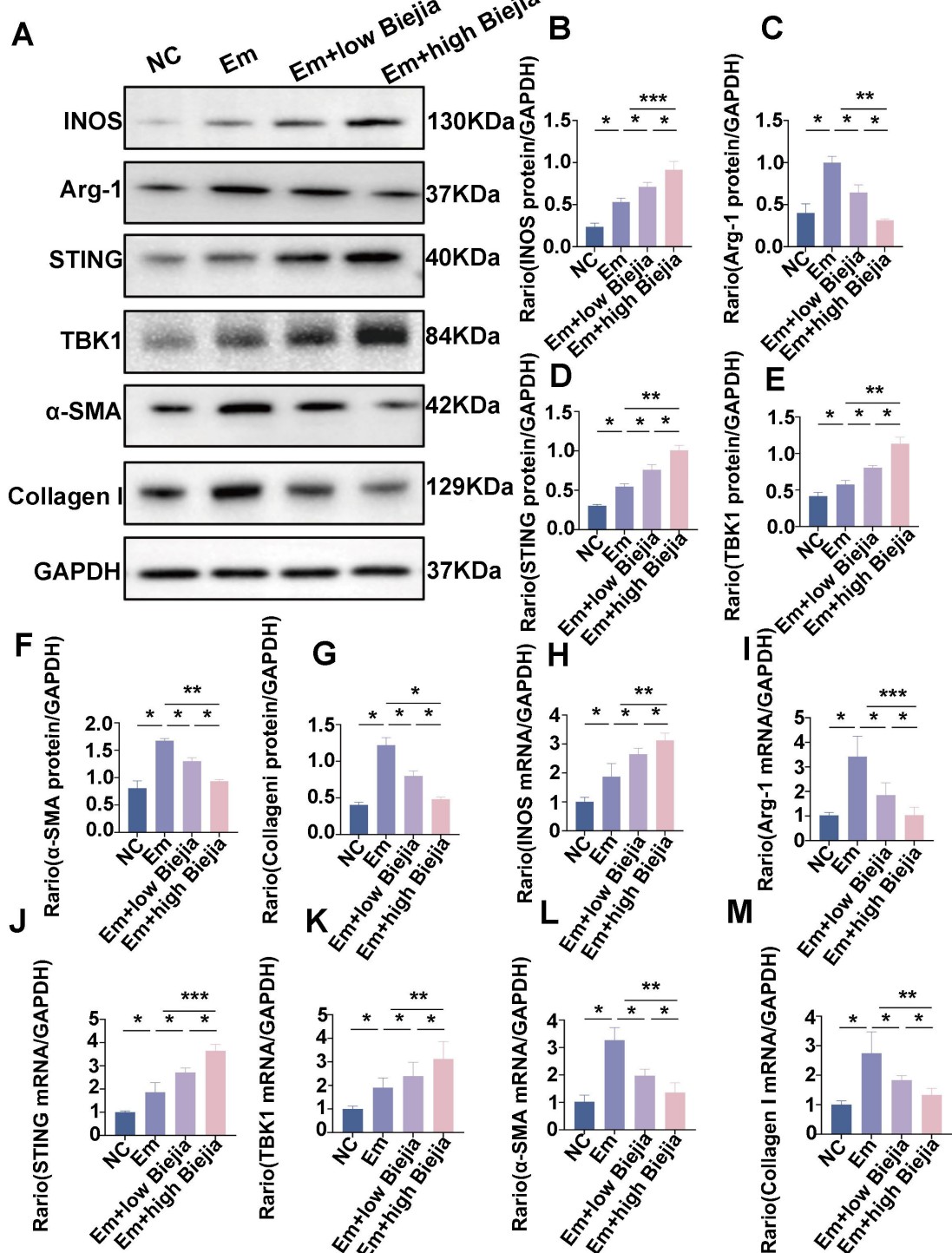

**FIG 4** Molecular profiling of STING-TBK1 pathway activation, macrophage polarization, and fibrosis markers in the liver of *E.m*. mice treated with CBRT. (A–G) Protein expression analysis by WB. (A) Representative Western blot images for iNOS, Arg-1, STING, TBK1, α-SMA, Collagen I, and GAPDH (loading control) in liver tissues from each group. (B–G) Densitometric quantification of the protein levels shown in (A). (H–M) Gene expression analysis by qRT-PCR. (H) mRNA expression levels of iNOS, Arg-1, STING, TBK1, α-SMA, Collagen I, and GAPDH (reference gene) in liver tissues. (I–M) Corresponding statistical analysis of the qRT-PCR data. Data are from three biologically independent replicates (*n* = 3) and presented as mean ± SEM. Statistical significance was determined using unpaired Student's *t*-test (for comparisons between two groups) or one-way ANOVA (for multi-group comparisons followed by appropriate post hoc tests), as indicated: *P < 0.05, **P < 0.01, and ***P < 0.001.

increased the expression of the M1 marker Inos (Fig. 4A, B and H) at both the protein and mRNA levels. We propose that CBRT alleviates liver fibrosis by activating the STING-TBK1 pathway to reprogram macrophage polarization, thereby counteracting the Th2/M2-dominated immunosuppressive microenvironment and partially restoring immune homeostasis. To assess the dynamic activation of the STING-TBK1 signaling pathway during *E.m.* infections, we performed IF analysis on liver tissues from infected mice. The pathway was activated by day 8 post-infection (Fig. S2A and B), reached peak activity at day 30 concomitant with heightened inflammation (Fig. S2C), and subsequently declined, as shown by progressively weaker STING-TBK1 co-localization at days 60 and 90 (Fig. S2D), indicating pathway suppression during chronic infection. CBRT intervention restored STING-TBK1 co-localization (Fig. S2E through I). Together with corroborative WB and qRT-PCR data, these results demonstrate that CBRT reactivates the STING-TBK1 pathway, inhibits the immunosuppressive M2 macrophage polarization, and alleviates hepatic fibrosis.

## Serum containing CBRT ameliorates hepatocyte fibrosis by activating the STING-TBK1 signaling pathway, which in turn suppresses the polarization of macrophages toward the M2 phenotype

Based on the clinical dosage regimen of CBRT, three intervention protocols (T1, T2, and T3) with different concentrations and administration frequencies were designed for oral gavage in mice (Fig. 5A). Blood was collected from the retro-orbital venous plexus at predetermined time points. The inhibitory effects of the drug-containing sera, obtained following the different dosing frequencies, on mHSC proliferation are presented in (Fig. 5B). The corresponding areas under the curve (AUCs) for the three groups were 525.18 ± 33.62, 613.95 ± 24.87, and 575.97 ± 23.90, respectively. Statistical analysis revealed significant differences among the T1, T2, and T3 groups ($P < 0.05$). Based on the AUC results, the T2 protocol was selected as the high-dose CBRT intervention group, while the T3 protocol was designated as the low-dose intervention group. Based on these findings, we established the *in vitro* co-culture system and determined the high and low doses of CBRT for the subsequent *in vivo* mouse experiments (Fig. 5C). The results showed that after CBRT intervention in the alveolar hydatid infection microenvironment, the expressions of the main molecules STING and TBK1 proteins and mRNAs in the STING-TBK1 signaling pathway of GHA1 cells increased (Fig. 5G, H, M and N). RAW264.7 cells were polarized to M1, and the expressions of iNOS proteins and mRNAs increased (Fig. 5D, E and K), while the expressions of Arg-1 proteins and mRNAs decreased (Fig. 5D, F and L). An EmP-based microenvironment model of infection was established. Following the CBRT intervention, the following changes were observed: the protein and mRNA expression levels of components in the STING-TBK1 signaling pathway were upregulated (Fig. 5D, G, H, M and N). The polarization of macrophages toward the M2 phenotype was inhibited, as evidenced by decreased Arg-1 protein and mRNA expression (Fig. 5D, F and L). Concurrently, the expression of the M1 macrophage polarization marker iNOS was increased at both the protein and mRNA levels (Fig. 5D, E and K). Furthermore, in normal hepatocyte GAH1 cells, the expression of fibrosis markers α-SMA and Collagen I was reduced, as measured by both protein and mRNA analysis (Fig. 5D, I, J, O and P).

## Under the microenvironment established by EmP, CBRT demonstrated the ability to suppress hepatocyte apoptosis as well as macrophage invasion and migration

Under EmP-present conditions, cells were treated with either high- or low-dose CBRT. Flow cytometric apoptosis analysis revealed that CBRT could inhibit hepatocyte apoptosis (Fig. S1A and B) and also suppress the invasive and migratory capacities of RAW264.7 cells (Fig. S1C through F). These effects may represent an important mechanism through which CBRT alleviates *E.m.* infection-induced hepatocyte fibrosis.

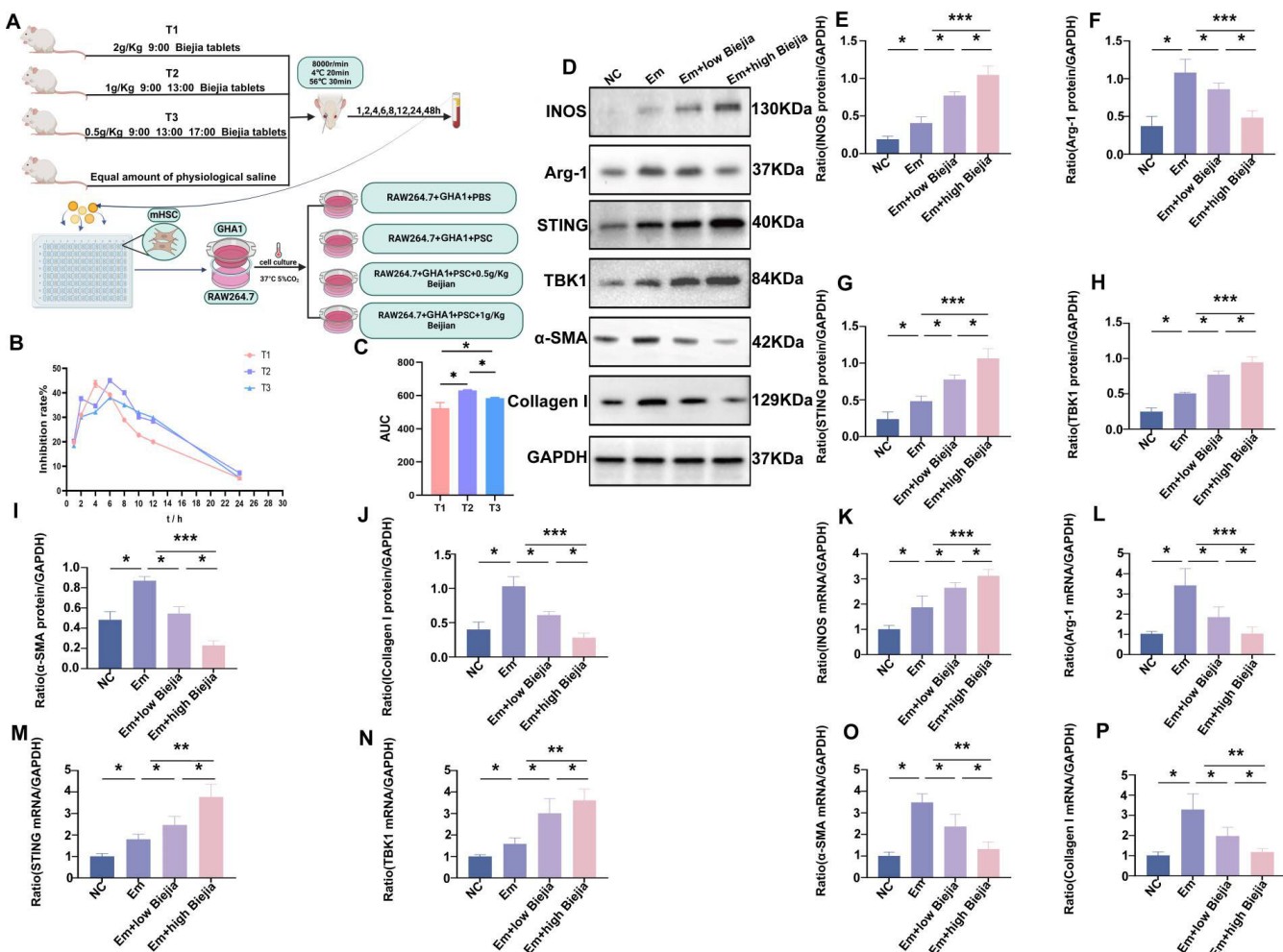

**FIG 5** Effects of drug-containing serum from CBRT-treated rats on hepatic stellate cells and the STING-TBK1 pathway in an *in vitro E.m.*-infected microenvironment. (A) Schematic diagram illustrating the establishment of the *in vitro E.m.*-infected co-culture model and the experimental grouping. (B and C) Effect of CBRT-containing serum on mouse mHSC viability. (B) Dose-response curve of mHSC inhibition rate treated with CBRT-containing serum (mean ± SEM, *n* = 5). (C) AUC analysis of the dose-response data shown in (B). (D–J) Protein expression analysis in the *in vitro*-infected microenvironment. (D) Representative Western blot images for iNOS, Arg-1, STING, TBK1, α-SMA, Collagen I, and GAPDH (loading control). (E–J) Densitometric quantification of the protein levels shown in (D). (K–P) mRNA expression levels of iNOS, Arg-1, STING, TBK1, α-SMA, Collagen I, and GAPDH (reference gene) as determined by qRT-PCR. Data are from three biologically independent replicates (*n* = 3) and presented as mean ± SEM. Statistical significance was determined using unpaired Student's *t*-test (for comparisons between two groups) or one-way ANOVA (for multi-group comparisons followed by appropriate post-hoc tests), as indicated: *$P < 0.05$, **$P < 0.01$, and ***$P < 0.001$.

## DISCUSSION

The liver is the primary organ affected by AE (1). This parasitic disease invades the liver, triggering inflammation and disrupting the normal hepatic architecture. In its early stages, AE progression is often insidious, yet it advances rapidly. The larval cysts of *E.m.* progressively invade and expand within the host liver, leading to fibrosis and eventual necrosis (2). Consequently, most patients are diagnosed at an advanced stage, after extensive intrahepatic metastasis has occurred, thereby missing the optimal window for curative surgical intervention (3). Therefore, exploring comprehensive treatment strategies for AE and identifying novel therapeutic targets to prevent or mitigate associated liver fibrosis are of critical clinical importance. The pathogenic mechanism of AE is complex. After the alveolar hydatid enters the host's liver, it first develops into vesicles, around which a granulomatous reaction occurs, fibrous connective tissue proliferates, and a large number of inflammatory cells (mainly macrophages) are recruited in the periphery. Fibroblasts accompanied by the activation of HSC secrete

extracellular matrices such as collagen to form fibrous tissue proliferation in a ring-like structure (14).

During AE infection, multiple immune cells are involved. Among these, Kupffer cells, the resident liver macrophages, play a central role. They produce various cytokines that can either promote or suppress inflammation (15). Based on their activation stimuli, macrophages are broadly classified into two major types. Classically activated (M1) macrophages, induced by cytokines such as IFN-γ, highly express iNOS. They exhibit potent antimicrobial and antiviral activity, constituting a crucial component of the host defense system (16). Alternatively activated (M2) macrophages, on the other hand, express Arg-1 (17). Studies have shown significant macrophage infiltration in the perilesional liver tissues of AE patients, with positive expression of M2 macrophage signature molecules. This suggests that M2 macrophages may be a key contributor to liver fibrosis in AE (5). These findings align with observations from the present study in liver lesions from advanced AE patients and *E.m.*-infected mice. They indicate that during the long-term chronic infection with the parasite, the functional heterogeneity of M2 macrophages—as pivotal immune participants—plays a significant role in shifting the host immune response from a pro-inflammatory to an anti-inflammatory dominant state, thereby further promoting hepatic fibrosis (5).

Previous studies have demonstrated the efficacy of CBRT in reversing liver fibrosis. For instance, a large-scale randomized controlled trial by Qu et al. (6) confirmed that CBRT can block and reverse hepatic fibrosis in patients with chronic hepatitis B or hepatocellular carcinoma. Similarly, Xu et al. (8) reported that CBRT, as an adjunct to entecavir, may benefit patients with chronic hepatitis B complicated by hepatic fibrosis. While these findings collectively establish the antifibrotic potential of CBRT, its specific effects on AE-induced liver fibrosis remain underexplored.

Guided by the clinically equivalent dose established in the drug instructions for CBRT, we evaluated the safety and efficacy of different dosages in an animal model. Administration of the key effective dose of 1 g/kg did not induce significant signs of acute toxicity in mice, as evidenced by unchanged general condition and stable serum biochemical markers of liver and kidney function (AST and ALT), indicating favorable tolerability. Moreover, treatment with this dose significantly reduced serum levels of AST, ALT, and key hepatic fibrosis markers in the model mice, demonstrating clear anti-fibrotic efficacy. Supported by the established long-term safety profile of CBRT from its approved clinical use for liver fibrosis in China, the preliminary safety observations in this study further validate the appropriateness of the selected experimental dose, particularly 1 g/kg. These findings provide a crucial dosage rationale and safety foundation for translating the pharmacological conclusions of this research into clinical practice. Furthermore, we observed that stimulation with high-dose CBRT-containing serum led to upregulated expression of the STING-TBK1 signaling pathway in cells. STING is an endoplasmic reticulum-resident dimeric adaptor protein, expressed in various hematopoietic cells, including macrophages, where it serves as a key regulator of type I interferon release and innate immunity (18). Its downstream kinase, TBK1, is essential for propagating STING-dependent signaling. Upon activation, STING translocates to the Golgi apparatus and recruits/activates TBK1 (19). Accumulating evidence implicates the STING-TBK1 axis as a pivotal modulator in fibrotic disorders (20). For example, Tryptanthrin was shown to promote hepatic inflammation and fibrosis by suppressing the cGAS-STING pathway and promoting M2 macrophage polarization (21). Conversely, Sun et al. (22) reported that Oroxylin A activates the STING pathway to promote IFN-β secretion and ameliorate liver fibrosis. These studies support the notion that STING-TBK1 activation can mitigate fibrosis, a conclusion aligned with our findings. However, contrasting evidence exists. Studies such as "Naringenin is a potential Immunomodulator for Inhibiting Liver Fibrosis by Inhibiting the cGAS-STING Pathway" (23) and "Expression of STING Is Increased in Liver Tissues from Patients with NAFLD and Promotes Macrophage-mediated Hepatic Inflammation and Fibrosis in Mice" (24) demonstrate that inhibiting this pathway can also alleviate fibrosis. This discrepancy suggests that the role of the STING-TBK1

pathway is context-dependent, exhibiting dualistic properties influenced by specific disease etiologies and activators—a complexity that warrants further investigation. Our study elucidates the context-dependent, and seemingly paradoxical, role of STING-TBK1 signaling in hepatic fibrosis. In sterile inflammatory models such as $CCl_4$-induced injury or NASH, persistent cellular damage leads to chronic overactivation of the STING pathway, driving deleterious inflammation and exacerbating fibrosis; thus, its inhibition represents a rational therapeutic strategy. In contrast, in the mid-to-late-stage *E.m*. infection model employed here, the liver establishes a strongly immunosuppressive and pro-fibrotic microenvironment dominated by a Th2/M2-type response. Under these conditions, our findings demonstrate that CBRT does not exacerbate injury but rather, through a "timely therapeutic enhancement" of the STING-TBK1 pathway, reprograms macrophages by inhibiting their polarization toward the M2 phenotype. This action alleviates immunosuppression, restores immune balance, and ultimately attenuates fibrosis. Consequently, we propose that the net effect of STING-TBK1 in liver fibrosis is context-dependent: it can act as a driver of excessive inflammation or as a lever to remodel the pathological microenvironment. Our results do not negate its pro-fibrotic role in sterile injury but establish that in infection-associated, immune-dysregulated hepatic fibrosis, this pathway can serve as a therapeutic target for achieving beneficial monoprogramming.

This study has certain limitations. While the findings of this study provide novel insights into the mechanisms by which CBRT intervenes in AE-induced liver fibrosis, we recognize that the adopted "drug-target-effect" model is relatively simplistic. AE fundamentally constitutes a multidimensional and dynamic ecosystem involving the host, parasite, gut microbiota, and the herbal components themselves. The gut microbiota may serve as a pivotal link mediating the therapeutic efficacy of CBRT. As an orally administered complex herbal formula, the bioavailability and metabolic transformation of CBRT's constituents are highly dependent on the functional state of the gut microbiota. Concurrently, parasitic infections are known to significantly alter the structure of the gut microbiota, thereby establishing a complex interaction network among CBRT, the gut microbiota, and the parasitic infection. Therefore, the systemic and hepatic effects observed in this study may partially originate from active metabolites generated by the gut microbiota. It must be noted that the current study did not perform comprehensive microbiome sequencing analysis of the gut microbiota. Future research employing integrated metagenomic and metabolomic analyses to decipher the dynamic changes along the "AE infection-gut microbiota-host metabolism" axis under CBRT intervention will enable a more precise delineation of its true pharmacologically active material basis and initial site of action. Secondly, regarding the dynamic nature of the parasitic niche, while this study confirms that CBRT can induce M1 macrophage polarization and alleviate fibrosis, the impact of this immune reprogramming on parasite survival requires deeper investigation. The fibrotic encapsulation represents a dynamic host-parasite interface that provides the parasite with both protection and nourishment. The M1-type response may suppress cyst growth through direct cytotoxic effects (e.g., via nitric oxide) or by indirectly disrupting its microenvironmental support (i.e., causing "metabolic starvation"). Future studies are needed to clarify the specific operative pathways.

BALB/c mice were selected based on their inherent Th2/M2-biased immune response, which promotes robust fibrous encapsulation following infection and leads to a pronounced, stable hepatic fibrotic phenotype—making them suitable for evaluating antifibrotic drug efficacy. Moreover, the *E.m*. infection model in BALB/c mice is internationally recognized and well-established, reliably recapitulating key features of human AE-related liver fibrosis. However, the immune microenvironment in BALB/c mice may favor M2 macrophage polarization, potentially accentuating the observed "M2-dominant fibrotic" phenotype in our model. Thus, the use of this strain might amplify the severity of M2-associated fibrosis, possibly rendering the CBRT-induced shift toward M1 polarization and subsequent attenuation of fibrosis more discernible. To assess the

generalizability of our findings, future studies will validate the therapeutic effects and mechanisms of CBRT in a C57BL/6 mouse model of AE. This comparative approach will provide a more comprehensive understanding of CBRT's immunomodulatory actions and strengthen the translational relevance of our conclusions. In summary, these findings support the potential applicability of CBRT in AE treatment. This work encourages further clinical-translational research aimed at alleviating various pathologies associated with liver fibrosis in AE patients.

## ACKNOWLEDGMENTS

This work was supported by the National Natural Science Foundation of China (82360127); the Key Project of Natural Science Foundation of Xinjiang Autonomous Region (2022D01D73); Tianshan Elite Talents Program for High-Level Medical and Health Professionals (TSYC202401B150); Xinjiang Uygur Autonomous Region Natural Science Foundation (General Program: 2023D01C155); Xinjiang Strategic Talent Cultivation Program: Specialized Talent Project for Urgent Needs (Project No. XJRC-2025-RS-PY-TX-032)

## AUTHOR AFFILIATIONS

[1]State Key Laboratory of Pathogenesis, Prevention and Treatment of High Incidence Diseases in Central Asia, Clinical Laboratory Center, Tumor Hospital Affiliated to Xinjiang Medical University, Urumqi, Xinjiang, China
[2]Medical Testing Center, The First Affiliated Hospital of Xinjiang Medical University, Urumqi, Xinjiang, China
[3]Medical Testing Center, Xinjiang Medical University Affiliated Traditional Chinese Medicine Hospital, Urumqi, Xinjiang, China
[4]Basic Medical College of Xinjiang Medical University, Urumqi, Xinjiang, China
[5]Medical Testing Center, The Fifth Affiliated Hospital of Xinjiang Medical University, Urumqi, Xinjiang, China

## AUTHOR ORCIDs

Xiumin Ma  http://orcid.org/0000-0001-8011-7513
Liang Wang  http://orcid.org/0009-0007-3697-3970

## AUTHOR CONTRIBUTIONS

Yuyu Ma, Writing – original draft | Hui Zhao, Data curation | Yumei Liu, Formal analysis | Menggen Meng, Data curation | Jiahui Chen, Data curation | Yuqin Sun, Formal analysis | Bin Fan, Data curation | Xuan Zhou, Formal analysis | Fengmin Tian, Investigation | Xiumin Ma, Conceptualization | Liang Wang, Data curation

## ADDITIONAL FILES

The following material is available online.

Supplemental Material

**Fig. S1 (Spectrum02115-25-s0001.tif).** Effects of CBRT-containing serum on apoptosis of GHA1 cells and migration of RAW264.7 macrophages in an in vitro E.m. mode.
**Fig. S2 (Spectrum02115-25-s0002.tif).** Dynamic co-localization of STING and TBK1 in hepatic tissues during E.m. progression.
**Supplemental material (Spectrum02115-25-s0003.docx).** Supplemental figure legends.
**Table S1 (Spectrum02115-25-s0004.docx).** Manufacturer and article number of reagents used in research.

## Open Peer Review

**PEER REVIEW HISTORY (review-history.pdf).** An accounting of the reviewer comments and feedback.

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
