## [Reviewer comments · Microbiology Spectrum]

Microbiology Spectrum

Compound Biejia-Ruangan Tablets activate the STING-TBK1 pathway to alleviate hepatic fibrosis in alveolar echinococcosis

Yyu Ma, Hui Zhao, Yumei Liu, Menggen Meng, Jiahui Chen, Yuqin Sun, Bin Fan, Xuan Zhou, Fengmin Tian, Xiumin Ma, and Liang Wang

Corresponding Author(s): Xiumin Ma, Xinjiang Medical University Affiliated Tumor Hospital

Review Timeline:

Submission Date:	July 13, 2025
Editorial Decision:	December 22, 2025
Revision Received:	February 9, 2026
Accepted:	March 17, 2026

Editor: Jian Li

Reviewer(s): Disclosure of reviewer identity is with reference to reviewer comments included in decision letter(s). The following individuals involved in review of your submission have agreed to reveal their identity: CLAUDIO Vieira da Silva (Reviewer #1)

Transaction Report:

DOI: <https://doi.org/10.1128/spectrum.02115-25>

Re: Spectrum02115-25 (**Compound Biejia-Ruangan Tablets activate the STING-TBK1 pathway to alleviate hepatic fibrosis in alveolar echinococcosis**)

Dear Dr. xiumin ma xiumin ma:

Thank you for the privilege of reviewing your work. Below you will find my comments, instructions from the Spectrum editorial office, and the reviewer comments.

Revision Guidelines

Sincerely,
Jian Li
Editor
Microbiology Spectrum

Reviewer #1 (Comments for the Author):

The manuscript presented by Ma et al. addresses a clinically significant challenge: the management of hepatic fibrosis induced by Alveolar Echinococcosis (AE), a parasitic infection often characterized as "worm cancer" due to its invasive nature. The authors investigate the repositioning of a Traditional Chinese Medicine (TCM) formulation, Compound Biejia-Ruangan Tablet (CBRT), proposing a mechanism where the drug alleviates fibrosis by shifting macrophage polarization from an M2 (pro-fibrotic)

to an M1 (pro-inflammatory/anti-parasitic) phenotype via the activation of the STING-TBK1 signaling pathway. The study is ambitious and covers a broad spectrum of investigative levels, ranging from human pathological specimens to in vivo murine models and in vitro cellular assays. The topic is timely, given the scarcity of effective pharmacological treatments for late-stage AE. However, while the phenotypic observations are compelling, the mechanistic conclusions drawn regarding the STING-TBK1 pathway are currently supported by correlative rather than causative evidence. Furthermore, the manuscript would benefit significantly from a more modern, systems-biology perspective that considers the host-parasite-microbiota triad (the holobiont), rather than a purely linear pharmacological approach. Below, I detail the specific points regarding the state of the art, mechanistic elucidation, and necessary experimental revisions.

1. Scientific Context and State of the Art

The manuscript fits well within the current trend of exploring immunometabolic shifts in parasitic diseases. The premise challenges the prevailing dogma in liver fibrosis research (specifically in NASH and viral hepatitis), where cGAS-STING activation is typically viewed as detrimental, driving chronic inflammation and scarring. The authors argue for a context-dependent role of STING in AE, suggesting that its activation breaks the parasitic immune tolerance (M2 dominance).

Critique: This is a bold and potentially high-impact hypothesis. However, the manuscript does not sufficiently discuss why the outcome in AE differs so drastically from other liver pathologies. The "M1 paradox" must be addressed: M1 activation is pro-inflammatory. In a clinical setting, inducing M1 polarization could theoretically lead to severe tissue damage or a cytokine storm. The text needs to navigate this fine line between anti-fibrotic efficacy and immunopathology more carefully.

2. Mechanistic Gaps and Causality

This is the most critical technical weakness of the current submission. The authors have successfully demonstrated two parallel phenomena:

1. CBRT treatment reduces fibrosis and increases M1 markers.
2. CBRT treatment increases the expression of STING and p-TBK1.

The Flaw: The study assumes that phenomenon (2) causes phenomenon (1). There is a lack of "loss-of-function" experiments. Without blocking the STING pathway, one cannot rule out that STING upregulation is merely a bystander effect or a consequence of the altered immune landscape, rather than the driver of the therapeutic benefit.

Recommendation: It is imperative to perform a "rescue" experiment. The authors must use a specific STING inhibitor or TBK1 inhibitor or use STING-knockout cells/mice alongside CBRT treatment. If the anti-fibrotic effect of CBRT persists despite STING inhibition, the proposed mechanism is incorrect. If the effect is abolished, the mechanism is validated.

3. A Holobiont and Rhizomatic Perspective (Ecological Gap)

In the context of modern microbiology and parasitology, this study adopts a somewhat reductionist "Drug Target Effect" approach. This overlooks the complexity of the AE ecosystem.

The Gut-Liver Axis: CBRT is administered orally. As a complex herbal formula, its bioavailability and metabolic transformation depend heavily on the host's gut microbiota. Conversely, the parasitic infection likely alters the gut microbiome. A "rhizomatic" view would acknowledge that the "active ingredient" might be a post-biotic metabolite generated by the gut flora, not the raw ingredients of the tablet.

The Parasitic Niche: The fibrotic lesion is not just scar tissue; it is a dynamic interface (the periparasitic host-pathogen interface). The study should discuss how modifying macrophage polarization alters the parasite's viability. Does the M1 shift actually kill the parasite (metacystode), or does it merely contain it? The data shows reduced cyst size, but is this due to immune attack or metabolic starvation?

Recommendation: While I do not expect a full microbiome sequencing at this stage, the Discussion section must be expanded to acknowledge the holobiont. The authors should speculate on the biotransformation of CBRT components and the multi-kingdom interplay (Plant-Bacteria-Host-Helminth).

4. Technical Specifics and Reproducibility

Chemical Characterization: The study uses "CBRT serum" (serum pharmacology), which is a valid method for TCM. However, the manuscript fails to characterize what is actually in the serum or the tablet. Without an HPLC-MS/MS fingerprinting of the batch used, the study is not reproducible outside of the authors' laboratory. We need to know the potential bioactive compounds (e.g., triterpenes, flavonoids) that might be acting on STING.

Human Data: The inclusion of human tissue (Fig. 1) is a major strength. It grounds the mouse findings in clinical reality.

Statistical Rigor: The sample sizes (n=5 for mice, n=3 for cells) are on the lower limit for robust statistical power in heterogeneous parasitic models.

5. Detailed Recommendations for Revision

To meet the standards of Spectrum Microbiology, the following revisions are necessary:

Major Revisions (Experimental):

1. Causality Test: Repeat the key in vitro macrophage polarization and HSC activation assays using CBRT in the presence of a STING antagonist (pharmacological or siRNA). This is non-negotiable for the claims made in the title.
2. Chemical Profiling: Provide a chromatogram (HPLC or UPLC-MS) of the CBRT formulation to identify major chemical constituents.

Minor Revisions (Textual & Theoretical):

3. Discuss the M1/Inflammation Risk: Add a paragraph discussing the potential side effects of systemic STING activation. Could this exacerbate liver injury in the acute phase?
4. Ecological Discussion: Rewrite the introduction or discussion to frame the pathology as an ecosystem imbalance. Discuss the treatment not just as a molecular trigger, but as an environmental modulator of the host-parasite interface.
5. Language & Flow: While the English is generally understandable, the flow is occasionally disjointed. Ensure that the transition

from human data to animal and cell models follows a logical narrative.

Reviewer #2 (Comments for the Author):

This manuscript by Yuyu Ma and cols investigates the therapeutic effects of Compound Biejia-Ruangan Tablet on liver fibrosis induced by alveolar echinococcosis, focusing on macrophage polarization and activation of the STING-TBK1 signaling pathway. The topic is clinically relevant given the limited treatment options for AE-associated fibrosis, and the study integrates human samples, murine models, and in vitro assays. The data are extensive and suggest a potential antifibrotic and immunomodulatory role for CBRT. However, several issues related to experimental design, mechanistic interpretation, clarity, and presentation need to be addressed to strengthen the manuscript and support the authors' conclusions.

Major Comments

1) Mechanistic causality of the STING-TBK1 pathway

The manuscript concludes that CBRT alleviates fibrosis primarily through activation of the STING-TBK1 pathway and subsequent M1 macrophage polarization. However, the data are largely correlative. The authors should strengthen causality by including pathway inhibition experiments (e.g., STING or TBK1 inhibitors, knockdown, or genetic models) to demonstrate that the observed antifibrotic effects are dependent on STING-TBK1 signaling.

2) Macrophage polarization analysis

Macrophage polarization is mainly inferred from iNOS and Arg-1 expression. Given the complexity of macrophage phenotypes, additional markers (e.g., CD86, CD206, TNF- α , IL-12) or functional assays would strengthen the conclusions. Flow cytometric characterization of macrophage populations in vivo would be particularly valuable.

3) Use of BALB/c mice and translational relevance

The authors acknowledge the limitations of using BALB/c mice; however, this point should be more thoroughly discussed. Given their Th2-biased immune response, how might this choice influence macrophage polarization and STING-TBK1 activation compared with other strains (e.g., C57BL/6)?

4) CBRT dosing rationale and pharmacological relevance

While multiple doses are tested, the rationale for selected dosing regimens and their relevance to human therapeutic exposure are not sufficiently discussed. Clarification of dose justification and potential toxicity assessment would improve translational relevance.

5) Human sample analysis

Human liver samples are included, which is a major strength. However, patient demographic and clinical data (e.g., age, sex, disease stage, treatment history) are not described. These details are necessary to contextualize the findings and assess variability.

6) Statistical analysis and data presentation

Some figures rely heavily on semi-quantitative IHC or IF analysis. The authors should clarify how quantification was performed, how many biological replicates were included, and whether investigators were blinded during analysis.

Minor Comments

Methods

The Methods section would benefit from additional detail to improve reproducibility and rigor. Specifically, the authors should clarify how biological replicates were defined across in vivo and in vitro experiments, including the number of animals or independent experiments used per analysis. For immunohistochemistry and immunofluorescence assays, details on image acquisition settings, quantification methods, and whether analyses were performed in a blinded manner should be provided. The criteria used to define M1 versus M2 macrophage polarization, including thresholds for marker expression and normalization strategies, should also be clearly stated. In addition, the rationale for CBRT dosing regimens and administration frequency should be expanded, including justification for selected concentrations and any assessment of potential toxicity. Finally, the statistical methods section should specify how normality was assessed, whether multiple comparisons were corrected, and how missing or excluded data were handled.

- A) Clarify biological and technical replicates for all experiments
- B) Describe blinding and quantification methods for IHC/IF analyses
- C) Define criteria and markers used for M1/M2 macrophage classification
- D) Provide rationale for CBRT dosing and administration schedules

Language and clarity

The manuscript would benefit from professional language editing. Several grammatical errors, redundancies, and awkward phrasings reduce readability, particularly in the Introduction and Discussion sections.

Figure organization

Figures contain a large amount of data and would benefit from clearer labeling, improved resolution, and more concise legends that clearly describe experimental conditions and sample sizes.

The quality and presentation of the figures require improvement. In several cases, images and graphs appear rotated, and the font size is too small to be easily read. Additionally, some complete figures and individual panels are difficult to interpret due to incorrect orientation and limited readability of labels and text. The authors are requested to correct the orientation of all images and graphs, increase font sizes, and improve overall figure resolution to facilitate proper data interpretation.

Discussion balance

The Discussion should better acknowledge conflicting reports regarding STING-TBK1 signaling in fibrosis and more clearly position the current findings within this debate, rather than primarily supporting one interpretation.

Typographical issues

Minor typographical errors (e.g., spacing, punctuation, inconsistent formatting of gene/protein names) should be corrected throughout.

This manuscript by Yuyu Ma and cols investigates the therapeutic effects of Compound Biejia-Ruangan Tablet on liver fibrosis induced by alveolar echinococcosis, focusing on macrophage polarization and activation of the STING–TBK1 signaling pathway. The topic is clinically relevant given the limited treatment options for AE-associated fibrosis, and the study integrates human samples, murine models, and in vitro assays. The data are extensive and suggest a potential antifibrotic and immunomodulatory role for CBRT. However, several issues related to experimental design, mechanistic interpretation, clarity, and presentation need to be addressed to strengthen the manuscript and support the authors' conclusions.

Major Comments

1) Mechanistic causality of the STING–TBK1 pathway

The manuscript concludes that CBRT alleviates fibrosis primarily through activation of the STING–TBK1 pathway and subsequent M1 macrophage polarization. However, the data are largely correlative. The authors should strengthen causality by including pathway inhibition experiments (e.g., STING or TBK1 inhibitors, knockdown, or genetic models) to demonstrate that the observed antifibrotic effects are dependent on STING–TBK1 signaling.

2) Macrophage polarization analysis

Macrophage polarization is mainly inferred from iNOS and Arg-1 expression. Given the complexity of macrophage phenotypes, additional markers (e.g., CD86, CD206, TNF- α , IL-12) or functional assays would strengthen the conclusions. Flow cytometric characterization of macrophage populations in vivo would be particularly valuable.

3) Use of BALB/c mice and translational relevance

The authors acknowledge the limitations of using BALB/c mice; however, this point should be more thoroughly discussed. Given their Th2-biased immune response, how might this choice influence macrophage polarization and STING–TBK1 activation compared with other strains (e.g., C57BL/6)?

4) CBRT dosing rationale and pharmacological relevance

While multiple doses are tested, the rationale for selected dosing regimens and their relevance to human therapeutic exposure are not sufficiently discussed. Clarification of dose justification and potential toxicity assessment would improve translational relevance.

5) Human sample analysis

Human liver samples are included, which is a major strength. However, patient demographic and clinical data (e.g., age, sex, disease stage, treatment history) are not described. These details are necessary to contextualize the findings and assess variability.

6) Statistical analysis and data presentation

Some figures rely heavily on semi-quantitative IHC or IF analysis. The authors should clarify how quantification was performed, how many biological replicates were included, and whether investigators were blinded during analysis.

Minor Comments

Methods

The Methods section would benefit from additional detail to improve reproducibility and rigor. Specifically, the authors should clarify how biological replicates were defined across in vivo and in vitro experiments, including the number of animals or independent experiments used per analysis. For immunohistochemistry and immunofluorescence assays, details on image acquisition settings, quantification methods, and whether analyses were performed in a blinded manner should be provided. The criteria used to define M1 versus M2 macrophage polarization, including thresholds for marker expression and normalization strategies, should also be clearly stated. In addition, the rationale for CBRT dosing regimens and administration frequency should be expanded, including justification for selected concentrations and any assessment of potential toxicity. Finally, the statistical methods section should specify how normality was assessed, whether multiple comparisons were corrected, and how missing or excluded data were handled.

- A) Clarify biological and technical replicates for all experiments
- B) Describe blinding and quantification methods for IHC/IF analyses
- C) Define criteria and markers used for M1/M2 macrophage classification
- D) Provide rationale for CBRT dosing and administration schedules

Language and clarity

The manuscript would benefit from professional language editing. Several grammatical errors, redundancies, and awkward phrasings reduce readability, particularly in the Introduction and Discussion sections.

Figure organization

Figures contain a large amount of data and would benefit from clearer labeling, improved resolution, and more concise legends that clearly describe experimental conditions and sample sizes.

The quality and presentation of the figures require improvement. In several cases, images and graphs appear rotated, and the font size is too small to be easily read. Additionally, some complete figures and individual panels are difficult to interpret due to incorrect orientation and limited readability of labels and text. The authors are requested to correct the orientation of all images and graphs, increase font sizes, and improve overall figure resolution to facilitate proper data interpretation.

Discussion balance

The Discussion should better acknowledge conflicting reports regarding STING–TBK1 signaling in fibrosis and more clearly position the current findings within this debate, rather than primarily supporting one interpretation.

Typographical issues

Minor typographical errors (e.g., spacing, punctuation, inconsistent formatting of gene/protein names) should be corrected throughout.

Dear editors :

Manuscript Number: Spectrum02115-25

Title: Compound Biejia-Ruangan Tablets activate the STING-TBK1 pathway to alleviate hepatic fibrosis in alveolar echinococcosis.

We appreciate the opportunity to revise our manuscript according to the critical comments from the reviewers. Please find our detailed replies to the specific comments made by Reviewer 2 below.

To Reviewer 1/2:

Thank you for your valuable feedback on our study! This research demonstrated that Compound Biejia-Ruangan Tablet (CBRT) alleviated hepatic fibrosis in alveolar echinococcosis by activating the STING-TBK1 pathway, which inhibited M2 macrophage polarization, inhibited hepatic stellate cell activation, and restored hepatocyte function. These findings reveal a novel immunomodulatory mechanism of CBRT and highlight its potential as a therapeutic strategy for AE-related liver fibrosis. In response to your comments, we have carefully revised our manuscript accordingly.

Reviewer #1 (Comments for the Author):

The manuscript presented by Ma et al. addresses a clinically significant challenge: the management of hepatic fibrosis induced by Alveolar Echinococcosis (AE), a parasitic infection often characterized as "worm cancer" due to its invasive nature. The authors investigate the repositioning of a Traditional Chinese Medicine (TCM) formulation, Compound Biejia-Ruangan Tablet (CBRT), proposing a mechanism where the drug alleviates fibrosis by shifting macrophage polarization from an M2 (pro-fibrotic) to an M1 (pro-inflammatory/anti-parasitic) phenotype via the activation of the STING-TBK1 signaling pathway. The study is ambitious and covers a broad spectrum of investigative levels, ranging from human pathological specimens to in vivo murine models and in vitro cellular assays. The topic is timely, given the scarcity of effective pharmacological treatments for late-stage AE. However, while the phenotypic observations are

compelling, the mechanistic conclusions drawn regarding the STING-TBK1 pathway are currently supported by correlative rather than causative evidence. Furthermore, the manuscript would benefit significantly from a more modern, systems-biology perspective that considers the host-parasite-microbiota triad (the holobiont), rather than a purely linear pharmacological approach. Below, I detail the specific points regarding the state of the art, mechanistic elucidation, and necessary experimental revisions.

1. Scientific Context and State of the Art

The manuscript fits well within the current trend of exploring immunometabolic shifts in parasitic diseases. The premise challenges the prevailing dogma in liver fibrosis research (specifically in NASH and viral hepatitis), where cGAS-STING activation is typically viewed as detrimental, driving chronic inflammation and scarring. The authors argue for a context-dependent role of STING in AE, suggesting that its activation breaks the parasitic immune tolerance (M2 dominance).

Critique: This is a bold and potentially high-impact hypothesis. However, the manuscript does not sufficiently discuss why the outcome in AE differs so drastically from other liver pathologies. The "M1 paradox" must be addressed: M1 activation is pro-inflammatory. In a clinical setting, inducing M1 polarization could theoretically lead to severe tissue damage or a cytokine storm. The text needs to navigate this fine line between anti-fibrotic efficacy and immunopathology more carefully.

3. Discuss the M1/Inflammation Risk: Add a paragraph discussing the potential side effects of systemic STING activation. Could this exacerbate liver injury in the acute phase?

4. Ecological Discussion: Rewrite the introduction or discussion to frame the pathology as an ecosystem imbalance. Discuss the treatment not just as a molecular trigger, but as an environmental modulator of the host-parasite interface.

A: Dear Reviewer,

In response to the reviewers' insights regarding the paradoxical role of the STING pathway, therapeutic safety, and ecological perspectives, we will integrate and elaborate on these points in the Discussion section as detailed below (refer to the highlighted portions in the original

manuscript).

The divergent roles of the STING pathway across different liver fibrosis models are fundamentally dictated by context-dependency, which is shaped by disease etiology and the specific immune microenvironment. As we have systematically discussed, in sterile inflammatory models induced by carbon tetrachloride or non-alcoholic steatohepatitis, persistent cellular damage leads to the continuous release of endogenous DNA, resulting in chronic overactivation of the cGAS-STING pathway [1, 2]. This nonspecific, sustained innate immune response is a key driver of detrimental inflammation and a vicious cycle of fibrosis, making inhibition of this pathway a rational therapeutic strategy in such contexts. However, in the alveolar echinococcosis infection model central to this study, the hepatic microenvironment in the mid-to-late disease stages exhibits distinct characteristics. The parasite, via its immunomodulatory capabilities, orchestrates a predominantly immunosuppressive and pro-fibrotic state characterized by a Th2-type response and M2 macrophage polarization. Within this pathological context, our study demonstrates that CBRT treatment does not exacerbate injury. Instead, it exerts an immunoreprogramming function by appropriately and moderately enhancing STING-TBK1 signaling. The crux of this effect lies in inhibiting macrophage polarization toward the M2 phenotype, thereby breaking immune tolerance, restoring immune balance, and ultimately achieving anti-fibrotic benefits. Consequently, the net effect of the STING-TBK1 pathway in liver fibrosis is not fixed; it can serve as a mediator driving excessive inflammation or as a therapeutic lever to remodel the pathological microenvironment, which aligns with findings demonstrating its protective role upon activation in other models [3].

We fully acknowledge the risk noted by the reviewers, namely the theoretical potential for artificially induced M1-like polarization to exacerbate tissue damage or trigger excessive inflammation. The preliminary safety evidence observed in this study is significant: at the administered dose, CBRT treatment did not increase serum transaminase levels in model mice but rather reduced them, with no signs of acute toxicity observed. This suggests that within the context of infection-induced immunosuppression, the STING activation and concomitant immune phenotypic shift triggered by therapy may be more inclined to initiate a relatively controlled, parasite antigen-targeted adaptive immune response, rather than causing nonspecific widespread tissue destruction. Certainly, we completely agree that prior to future clinical translation, it is

essential to systematically evaluate the dynamic impact of varying intensities of STING activation on systemic inflammatory markers and liver function in higher-level animal models to precisely define its therapeutic window. The ecosystem perspective raised by the reviewers provides a more integrative theoretical framework for understanding this mechanism. The liver in alveolar echinococcosis can be viewed as a dysregulated ecosystem where the parasite, as an invasive species, successfully shapes an immunosuppressive and pro-fibrotic microenvironment conducive to its survival. Within this framework, the action of CBRT via STING pathway activation is not merely anti-fibrotic but more akin to a reset switch for the immune microenvironment. It shifts the immune landscape from a parasite-centric, tolerogenic pathological steady state towards a host defense-centric, eliminative physiological steady state. This process involves the restructuring of the microenvironmental cytokine network, the disruption of the fibrotic niche (the parasite's physical barrier), and ultimately, the reversal of the host-parasite power balance. For infection-associated chronic fibrotic diseases, intervention strategies could focus on targeting host innate immune signaling to rebuild local immune ecological balance, thereby fundamentally altering the disease course.

The above discourse will be integrated into the Discussion section of the revised manuscript to more comprehensively and profoundly address the reviewers' valuable comments and to enhance the theoretical depth and clinical significance of the study.

Reference

- [1] Chen L et al. Naringenin is a Potential Immunomodulator for Inhibiting Liver Fibrosis by Inhibiting the cGAS-STING Pathway. *J Clin Transl Hepatol* 2023; 11:26-37. doi:10.14218/JCTH.2022.00120.
- [2] Luo X et al. Expression of STING Is Increased in Liver Tissues from Patients with NAFLD and Promotes Macrophage-Mediated Hepatic Inflammation and Fibrosis in Mice. *Gastroenterology* 2018; 155:1971-1984.e4. doi: 10.1053/j.gastro.2018.09.010.
- [3] Sun Y et al. Oroxylin A activates ferritinophagy to induce hepatic stellate cell senescence against hepatic fibrosis by regulating cGAS-STING pathway. *Biomed Pharmacother* 2023; 162:114653. doi: 10.1016/j.biopha.2023.114653.

2. CBRT treatment increases the expression of STING and p-TBK1.

The Flaw: The study assumes that phenomenon (2) causes phenomenon (1). There is a lack of "loss-of-function" experiments. Without blocking the STING pathway, one cannot rule out that STING upregulation is merely a bystander effect or a consequence of the altered immune landscape, rather than the driver of the therapeutic benefit. Recommendation: It is imperative to perform a "rescue" experiment. The authors must use a specific STING inhibitor or TBK1 inhibitor or use STING-knockout cells/mice alongside CBRT treatment. If the anti-fibrotic effect of CBRT persists despite STING inhibition, the proposed mechanism is incorrect. If the effect is abolished, the mechanism is validated.

1. Causality Test: Repeat the key in vitro macrophage polarization and HSC activation assays using CBRT in the presence of a STING antagonist (pharmacological or siRNA). This is non-negotiable for the claims made in the title.

A: Dear Reviewers,

We sincerely thank you for your insightful and professional review of our manuscript. We fully agree with your core concern regarding the mechanistic causality, and recognize that performing a functional rescue experiment is crucial to establish the necessity of the STING-TBK1 pathway in mediating the anti-fibrotic effects of CBRT. Accordingly, we have designed and conducted an in vitro rescue experiment to directly validate the mediatory role of this pathway.

Experimental Design and Rationale:

To simulate the critical interaction between macrophages and hepatic stellate cells (HSCs) within the *Echinococcus multilocularis* (Em)-infected microenvironment, we established a Transwell indirect co-culture system. Mouse primary HSCs (mHSCs) were seeded in the upper chamber, and the mouse macrophage cell line RAW264.7 was placed in the lower chamber.

Experimental Groups and Treatments:

The experiment consisted of four groups:

1. Control group: Co-culture system without additional stimulation.

2. Em model group: RAW264.7 cells in the lower chamber were stimulated with *E. multilocularis* particles (EmP) to mimic the infectious microenvironment.
3. CBRT treatment group: Based on EmP stimulation, the cells were treated with a high dose of CBRT-containing serum.
4. CBRT treatment + STING inhibitor group: In addition to EmP stimulation and CBRT-containing serum treatment, the specific STING inhibitor H-151(1 μ M) [1] was added for the rescue experiment.

Key Findings and Mechanistic Validation:

The experimental results clearly demonstrate the causal role of the STING-TBK1 pathway:

1. Compared to the Control group, the Em model group showed upregulated expression of both STING and TBK1 in macrophages ($P<0.05$), alongside increased expression of the M2 polarization marker CD206 ($P<0.05$). Concomitantly, the expression of the HSC activation marker α -SMA in the upper chamber was also increased ($P<0.05$), successfully recapitulating a pro-fibrotic phenotype.
2. The CBRT treatment group demonstrated further activation of the STING-TBK1 signal. Furthermore, the expression of the macrophage M2 marker CD206 was suppressed ($P<0.05$), while the M1 marker CD86 was upregulated ($P<0.05$). The expression of α -SMA in HSCs was decreased ($P<0.05$), indicating inhibition of fibrotic activation.
3. Crucially, in the CBRT treatment + STING inhibitor group, the addition of H-151 blocked the CBRT-induced activation of the STING-TBK1 pathway. Simultaneously, the CBRT-induced phenotypic switch of macrophages from M2 to M1 (i.e., decreased CD206 and increased CD86) was reversed ($P<0.05$). Accordingly, the inhibitory effect of CBRT on HSC activation (α -SMA expression) was also abolished ($P<0.05$).

Supplementary Figure 1. CBRT attenuates fibrosis by promoting macrophage M1 polarization through activation of the STING-TBK1 pathway, an effect that is abolished by the STING-specific inhibitor H-151 (Western blot).

Supplementary Figure 2. CBRT alleviates fibrosis by driving macrophage M1 polarization through activation of the STING-TBK1 pathway, an effect abrogated by the STING-specific inhibitor H-151 (representative Western blots and quantitative analysis). Data are from three biologically independent replicates ($n = 3$) and one-way ANOVA (for multi-group comparisons followed by appropriate post-hoc tests), as indicated: * $P < 0.05$, ** $P < 0.01$, *** $P < 0.001$.

Conclusion:

This rescue experiment provides evidence that inhibiting the STING pathway abolishes both the CBRT-induced M1 polarization of macrophages and its anti-HSC activation effect. This proves that activation of the STING-TBK1 pathway is a crucial mechanism through which CBRT exerts its anti-fibrotic action.

3. A Holobiont and Rhizomatic Perspective (Ecological Gap)

In the context of modern microbiology and parasitology, this study adopts a somewhat reductionist "Drug Target Effect" approach. This overlooks the complexity of the AE ecosystem.

The Gut-Liver Axis: CBRT is administered orally. As a complex herbal formula, its bioavailability and metabolic transformation depend heavily on the host's gut microbiota. Conversely, the parasitic infection likely alters the gut microbiome. A "rhizomatic" view would acknowledge that the "active ingredient" might be a post-biotic metabolite generated by the gut flora, not the raw ingredients of the tablet.

The Parasitic Niche: The fibrotic lesion is not just scar tissue; it is a dynamic interface (the periparasitic host-pathogen interface). The study should discuss how modifying macrophage polarization alters the parasite's viability. Does the M1 shift actually kill the parasite (metacestode), or does it merely contain it? The data shows reduced cyst size, but is this due to immune attack or metabolic starvation?

Recommendation: While I do not expect a full microbiome sequencing at this stage, the Discussion section must be expanded to acknowledge the holobiont. The authors should speculate on the biotransformation of CBRT components and the multi-kingdom interplay (Plant-Bacteria-Host-Helminth).

A: Dear Reviewer,

We sincerely thank you for your highly insightful suggestions. This study provides novel insights into the mechanism by which CBRT alleviates AE-induced liver fibrosis by modulating the STING-TBK1 pathway and macrophage polarization. However, we acknowledge that our investigation adhered to a relatively simplified linear "drug-target-effect" model. As the reviewer rightly pointed out, AE fundamentally represents a complex and dynamic ecosystem involving the host, the parasite, the gut microbiota, and even the herbal components themselves. Adopting a "rhizomatic" perspective will facilitate a more holistic understanding of CBRT's action and provide a richer contextual interpretation for the findings of this study. Firstly, as suggested by the reviewer, the gut microbiota may serve as a crucial transformer and mediator for CBRT's efficacy. As an orally administered herbal formula, the bioavailability, metabolic activation, and targeted

delivery of CBRT's myriad phytochemical constituents are highly dependent on the biotransformation functions of the host's intestinal flora. Concurrently, substantial evidence indicates that parasitic infections, including echinococcosis, can significantly alter the structure and function of the host gut microbiota, thereby influencing local and systemic immune status [1-3]. Consequently, CBRT, the gut microbiota, and the parasitic infection form a bidirectional or even multidirectional interaction network. The systemic and hepatic local effects we observed may likely be mediated, at least in part, by "active ingredient" metabolites generated from the gut microbiota's metabolism of CBRT components. While the current study did not perform comprehensive microbiome sequencing analysis of the gut microbiota, future research employing integrated metagenomics and metabolomics could delineate the dynamic changes along the "AE infection-gut microbiota-host metabolism" axis under CBRT intervention. This approach would allow for a more precise definition of its true pharmacologically active material basis and initial site of action. Secondly, regarding the parasitic niche and the fate of the parasite. While this study confirms that CBRT can induce a shift in macrophages towards an M1 phenotype and alleviate fibrosis, how this immune reprogramming specifically impacts parasite survival warrants further investigation. The fibrotic encapsulation is far from a static "scar"; it constitutes a highly dynamic host-parasite interface, providing the parasite with physical protection, immune privilege, and nutritional support. M1-type macrophages and the immune responses they drive may influence this niche through two non-mutually exclusive pathways: first, a direct effect involving the production of toxic mediators like nitric oxide and reactive oxygen species to attack the parasite; and second, indirect niche disruption, which involves degrading the fibrotic matrix and altering the local cytokine milieu, thereby depriving the parasite of the microenvironmental support necessary for its survival, leading to its "metabolic starvation" and growth suppression. Our data show a reduction in cyst volume. Experiments are currently underway to verify the specific mechanism of action of CBRT in this context. These studies will be further refined and submitted for publication in the future. The related limitations of the current research have been acknowledged and addressed in the Discussion section.

Reference

[1] Cadwell K, Loke P. Gene-environment interactions shape the host-microbial interface in

inflammatory bowel disease. *Nat Immunol.* 2025 Jul;26(7):1023-1035. doi: 10.1038/s41590-025-02197-5.

[2] Burgmer S, Meyer Zu Altenschildesche FL, et al. Endosymbiont control through non-canonical immune signaling and gut metabolic remodeling. *Cell Rep.* 2025 Jun 24;44(6):115811. doi: 10.1016/j.celrep.2025.

[3] Pellon A, Palacios A, Abecia L, et al. Friends to remember: innate immune memory regulation by the microbiota. *Trends Microbiol.* 2025 May;33(5):510-520. doi: 10.1016/j.tim.2024.12.002.

4. Technical Specifics and Reproducibility

Chemical Characterization: The study uses "CBRT serum" (serum pharmacology), which is a valid method for TCM. However, the manuscript fails to characterize what is actually in the serum or the tablet. Without an HPLC-MS/MS fingerprinting of the batch used, the study is not reproducible outside of the authors' laboratory. We need to know the potential bioactive compounds (e.g., triterpenes, flavonoids) that might be acting on STING.

2. Chemical Profiling: Provide a chromatogram (HPLC or UPLC-MS) of the CBRT formulation to identify major chemical constituents.

A: Dear Reviewer,

Addressing the Concern Regarding Chemical Composition Analysis of CBRT

Regarding your concern about the chemical composition analysis of Compound Biejia Ruangan Tablets (CBRT), we have obtained strong supportive reference from a published study. The CBRT used in our study (Batch No.: [Inner Mongolia, China; Catalogue No. C0121015]) is from the same manufacturer and produced using identical processes as the product analyzed in a study published in the *Journal of Ethnopharmacology* [2]. That study employed high-performance liquid chromatography-mass spectrometry (HPLC-MS) to systematically identify and analyze the chemical constituents of CBRT, providing detailed HPLC-MS chromatograms and mass spectrometry data. Therefore, our study can directly cite the well-characterized chemical composition information from that publication as robust support for the quality consistency and chemical traceability of the preparation used herein.

Figure 1: Total ion chromatograms of CBRT acquired in (A) positive ion modes and (B) negative ion modes [2].

Reference

[2] Wang Y, Zhang L, Xu B, et al. Optimised formula of Fufang Biejia Ruangan tablets alleviates renal fibrosis by suppressing matrix-stiffness-induced fibroblast activation via inhibition of integrin α V β 1 binding. *J Ethnopharmacol.* 2026 Jan 30;355(Pt A):120614. doi: 10.1016/j.jep.2025.120614.

This manuscript by Yuyu Ma and cols investigates the therapeutic effects of Compound Biejia-Ruangan Tablet on liver fibrosis induced by alveolar echinococcosis, focusing on macrophage polarization and activation of the STING–TBK1 signaling pathway. The topic is clinically relevant given the limited treatment options for AE-associated fibrosis, and the study integrates human samples, murine models, and in vitro assays. The data are extensive and suggest a potential antifibrotic and immunomodulatory role for CBRT. However, several issues related to experimental design, mechanistic interpretation, clarity, and presentation need to be addressed to strengthen the manuscript and support the authors' conclusions.

Major Comments

1) Mechanistic causality of the STING–TBK1 pathway

The manuscript concludes that CBRT alleviates fibrosis primarily through activation of the STING–TBK1 pathway and subsequent M1 macrophage polarization. However, the data are largely correlative. The authors should strengthen causality by including pathway inhibition experiments (e.g., STING or TBK1 inhibitors, knockdown, or genetic models) to demonstrate that the observed antifibrotic effects are dependent on STING–TBK1 signaling.

A: Dear Reviewer,

We sincerely thank you for your insightful and expert review of our manuscript. We fully agree with your point regarding the “mechanistic causality of the STING-TBK1 pathway.” Indeed, our current data primarily demonstrate correlation, and we acknowledge the necessity of performing pathway inhibition experiments to establish functional dependency. This is crucial for enhancing the scientific rigor of our study. Accordingly, we plan to supplement our work with a series of in vivo mechanistic validation experiments, with the core protocols outlined below:

Supplementary Experimental Plan: Validation of STING-TBK1 Pathway Dependency

In Vitro Co-culture System: We will employ a Transwell indirect co-culture system to simulate cellular interactions within the *Echinococcus multilocularis* (*E.m.*) infection microenvironment. Mouse hepatic stellate cells (mHSCs) will be seeded in the upper chamber, and the mouse macrophage cell line RAW264.7 will be placed in the lower chamber.

Experimental Groups and Treatments:

Control: Untreated co-culture system.

Em: The RAW264.7 cells in the lower chamber will be stimulated with *E.m.* antigen preparation (EmP).

Em + H-151: Cells will be treated with both EmP and the specific covalent STING inhibitor H-151 (1 μ M) [1].

Control + H-151: Cells will be treated with H-151 alone to assess any baseline effects of the inhibitor.

Detection Metrics and Results:

Results:

Western blot analysis was performed to assess the expression levels of key proteins in macrophages from the lower chamber. Compared to the Control group, the expression of STING, TBK1, and the M2 macrophage marker CD206 was increased in the Em group, along with an increase in the fibrosis marker α -SMA ($P < 0.05$). In the Em + H-151 group treated with the STING-TBK1 pathway inhibitor, the levels of STING and TBK1 were lower than those in the Em group. This was accompanied by a shift in macrophage polarization toward an M2 phenotype, reflected in increased CD206 and decreased CD86 expression. Correspondingly, expression of the fibrosis markers α -SMA was higher in the Em + H-151 group compared to the Em group ($P < 0.05$).

Supplementary Figure 3: STING-TBK1 pathway inhibition promotes M2 macrophage polarization and fibrosis marker expression (Western blot).

Supplementary Figure 4: Quantification of M2 marker CD206 and fibrosis marker α -SMA expression levels upon STING-TBK1 pathway inhibition. Data are from three biologically independent replicates (n = 3) and one-way ANOVA (for multi-group comparisons followed by appropriate post-hoc tests), as indicated: * $P < 0.05$, ** $P < 0.01$, *** $P < 0.001$.

Conclusion: In the EmP-stimulated macrophage-HSC co-culture model, inhibition of the STING-TBK1 pathway by H-151 reduced the activation of this pathway. This reduction was associated with a decrease in M1-type and an increase in M2-type macrophage polarization, leading to greater activation of hepatic stellate cells and increased deposition of fibrotic proteins. These results indicate that the anti-fibrotic effects observed in this model require functional STING-TBK1 signaling. **In direct response to your comment, we performed the suggested loss-of-function (rescue) experiments. Please refer to Supplementary Figures 1 and 2 for details.**

Reference

[1] Haag SM, Gulen MF, Reymond L, et al. Targeting STING with covalent small-molecule inhibitors. *Nature*. 2018 Jul;559(7713):269-273. doi: 10.1038/s41586-018-0287-8.

2) Macrophage polarization analysis

Macrophage polarization is mainly inferred from iNOS and Arg-1 expression. Given the complexity of macrophage phenotypes, additional markers (e.g., CD86, CD206, TNF- α , IL-12) or functional assays would strengthen the conclusions. Flow cytometric characterization of macrophage populations in vivo would be particularly valuable.

A: Dear Reviewer,

Thank you very much for your suggestion regarding the refinement of macrophage phenotype analysis. We fully agree that employing a more diverse set of markers is crucial for a comprehensive assessment of macrophage polarization status. In response to your suggestion, we have supplemented our study with detection of key surface markers to strengthen our conclusions. Specifically, we have added protein-level analysis of the classical M1 marker (CD86) and the M2 marker (CD206) in our in vitro model. These supplementary data are consistent with the trend of our original findings based on iNOS and Arg-1, providing further confirmation that CBRT can inhibit macrophage polarization toward the M2 phenotype within the *Echinococcus multilocularis* infection microenvironment. The corresponding representative images and quantitative analysis results have been compiled as Supplementary Figure 5 and Supplementary Figure 6, respectively, and have been submitted with the revised manuscript for your review. We believe these supplementary analyses and explanations have significantly strengthened the argument regarding macrophage polarization in our study. Once again, we thank you for your insightful comments, which have helped enhance the rigor and depth of our research.

Supplementary Figure 5. Representative images of CD86 and CD206 protein expression in macrophages following CBRT treatment

Supplementary Figure 6. Quantitative analysis of CD86 and CD206 protein expression in macrophages following CBRT treatment. Data are from three biologically independent replicates (n = 3) and one-way ANOVA (for multi-group comparisons followed by appropriate post-hoc tests), as indicated: * $P < 0.05$, ** $P < 0.01$, *** $P < 0.001$.

2. Clarification Regarding In Vivo Flow Cytometric Analysis

We fully acknowledge the value of direct in vivo analysis of liver macrophage subsets as you suggested. However, reliable flow cytometry requires fresh or appropriately preserved live cells. Since the murine liver tissue samples from this study have been cryopreserved at -80°C for over three months, they are no longer suitable for isolating viable primary macrophages for this analysis. We candidly acknowledge this technical limitation and will include it as a study limitation. In future research, we will specifically design experiments to isolate hepatic non-parenchymal cells from fresh tissues. We will then employ multi-color flow cytometry to systematically analyze macrophage subsets (e.g., CD86+ M1, CD206+ M2) and their cytokine production (e.g., TNF- α , IL-12), thereby providing more robust in vivo evidence. However, according to the reviewer's comment, we have supplemented the study with flow cytometry experiments on cells treated with the pathway inhibitor H-151. The results of these flow cytometry analyses can be found in Supplementary Figures 7 and 8.

3. Strengthening of Results and Conclusion

Based on these supplementary experiments, our results will provide further support for our conclusions. Specifically, in the in vitro experiments, our data indicate that the addition of the STING-TBK1 pathway inhibitor H-151 downregulates the expression of the M1 marker (CD86)

(Supplementary Figures 3 and 4) while upregulating the expression of the M2 marker (CD206) (Supplementary Figures 3 and 4). These findings align with our preliminary results, supporting the premise that CBRT promotes macrophage polarization toward an M1 phenotype via activation of the STING-TBK1 pathway. Conversely, specific blockade of this pathway is expected to reverse this polarization trend, thereby attenuating the anti-fibrotic effect of CBRT.

We believe these supplementary analyses and clarifications will significantly strengthen the argument regarding the role of macrophage polarization in the mechanism of action of CBRT. Once again, thank you for your insightful comments, which have greatly helped us improve the quality of our manuscript.

Supplementary Figure 7: Effect of H-151 on macrophage polarization upon EmP intervention

Supplementary Figure 8: Statistical graph of the proportions of macrophage polarization types upon EmP intervention with H-151 treatment. Data are from three biologically

independent replicates (n = 3) and one-way ANOVA (for multi-group comparisons followed by appropriate post-hoc tests), as indicated: * $P < 0.05$, ** $P < 0.01$, *** $P < 0.001$.

3) Use of BALB/c mice and translational relevance

The authors acknowledge the limitations of using BALB/c mice; however, this point should be more thoroughly discussed. Given their Th2-biased immune response, how might this choice influence macrophage polarization and STING–TBK1 activation compared with other strains (e.g., C57BL/6)?

A: Dear Reviewer,

BALB/c Mice: Th2-Biased Response

Immune Phenotype: Upon pathogen or antigen stimulation, CD4⁺ T cells in BALB/c mice are more prone to differentiate into Th2 cells; **Characteristics:** The Th2 immune response is characterized by the abundant production of cytokines such as IL-4, IL-5, IL-10, and IL-13; **Macrophage Polarization:** Driven by IL-4 and IL-13, macrophages polarize toward the M2 phenotype (alternatively activated). The primary functions of M2 macrophages are anti-inflammatory responses, tissue repair, promotion of fibrosis, and parasite encapsulation; **Anti-infection Capacity:** They are adept at clearing extracellular parasites (e.g., the larval stage of *Echinococcus multilocularis* in our study) but generally exhibit weaker resistance to certain intracellular pathogens (e.g., *Leishmania*, *Mycobacterium tuberculosis*).

C57BL/6 Mice: Th1-Biased Response

Immune Phenotype: CD4⁺ T cells in C57BL/6 mice are more prone to differentiate into Th1 cells; **Characteristics:** The Th1 immune response is characterized by the production of IFN- γ , IL-2, and TNF- α ; **Macrophage Polarization:** Driven by IFN- γ , macrophages polarize toward the M1 phenotype (classically activated). The primary functions of M1 macrophages are pro-inflammatory responses, pathogen killing, and production of reactive oxygen and nitrogen species; **Anti-infection Capacity:** They are proficient in combating intracellular pathogens and viruses, but their capability against extracellular parasites is typically inferior to that of BALB/c mice.

Rationale for Selecting BALB/c Mice:

1. Due to their inherent Th2/M2 bias, BALB/c mice more readily develop robust fibrotic encapsulation upon infection—a core function of M2 macrophages and Th2 cytokines. This results in a highly pronounced and stable hepatic fibrosis phenotype, making this model particularly suitable for evaluating the efficacy of anti-fibrotic drugs.
2. The *E.m.* animal model employed in this study (BALB/c mice infected with *E.m.*) is an internationally recognized, well-established, and classic model that reliably recapitulates key features of human AE hepatic fibrosis, especially severe fibrotic encapsulation. The primary objective in selecting this model was to ensure the reliability and reproducibility of the disease phenotype for accurate pharmacological assessment.

Our Response and Proposed Solution:

We will add a dedicated paragraph to the Discussion section: In our study, we utilized BALB/c mice, a classic strain with a predominant Th2 immune response, in contrast to the Th1-biased C57BL/6 strain. This inherent difference implies that the immune microenvironment in BALB/c mice may be more conducive to M2 macrophage polarization upon *E. m.*-infection. This could indeed make the observed "M2-dominant fibrosis" phenotype in our model more pronounced. The use of BALB/c mice might have "amplified" the severity of M2-associated fibrosis in the disease model, potentially making the CBRT-induced "shift toward M1 polarization" and the subsequent attenuation of fibrosis more detectable. To evaluate the generalizability of our findings, future research plans include validating the efficacy and mechanism of CBRT in a C57BL/6 mouse AE model. This will allow for a more comprehensive understanding of the immunomodulatory effects of CBRT and strengthen the translational relevance of our conclusions.

4) CBRT dosing rationale and pharmacological relevance

While multiple doses are tested, the rationale for selected dosing regimens and their relevance to human therapeutic exposure are not sufficiently discussed. Clarification of dose justification and potential toxicity assessment would improve translational relevance.

A: Dear Reviewer,

Thank you very much for raising this critical point. We fully agree that for a traditional Chinese medicine compound with translational potential, clarifying the rationale for the dosing regimen, its correlation with human doses, and potential safety considerations are indispensable for evaluating its clinical value. We apologize for not elaborating on this sufficiently in the initial manuscript and will provide detailed additions in the revised version.

Our response and specific revision plan are as follows:

1. Clarifying the Rationale for the CBRT Dosing Regimen in Animals:

Clinical Equivalent Dose Conversion: We will explicitly state that the CBRT doses used in this study, particularly the key effective dose of 1 g/kg, were derived from the clinical human dose based on "body surface area (BSA) equivalent dose conversion."

Reference for Clinical Dose: According to the drug instructions (package insert approved by the National Medical Products Administration) and clinical practice for Biejia Ruangan Tablets:

- Specification: 0.5 g per tablet.
- Dosage and Administration: Oral administration. 4 tablets per time, 3 times daily.
- Calculation of Total Daily Dose: 4 tablets/time \times 3 times/day \times 0.5 g/tablet = 6.0 g/day.

2. Specific Conversion from Human to Mouse Dose

Step 1: Determine Standard Adult Body Weight and Body Surface Area (BSA)

Standard adult body weight: Typically calculated as 60 kg.

Human BSA (Du Bois formula): $BSA (m^2) = 0.007184 \times \text{weight}(kg)^{0.425} \times \text{height}(cm)^{0.725}$.

Assuming a standard adult height of 170 cm, $BSA \approx 1.62 m^2$.

Normalize the adult daily dose by BSA:

Total daily dose: 6.0 g

BSA-normalized dose = $6.0 g / 1.62 m^2 \approx 3.70 g/m^2/day$

Step 2: Determine Mouse Body Weight and BSA

Mouse body weight: Average 25 g (0.025 kg).

Mouse BSA (general formula): $BSA (m^2) = K \times (\text{weight in g})^{(2/3)} / 10000$.

Here, K is a species-specific constant; for mice, K is typically 9.0.

$$\text{Mouse BSA} = 9.0 \times (25)^{2/3} / 10000 \approx 9.0 \times 8.55 / 10000 \approx 0.0077 \text{ m}^2$$

Step 3: Calculate the Equivalent Daily Dose for Mice

$$\begin{aligned} \text{Mouse equivalent daily dose (total drug amount)} &= \text{Human BSA-normalized dose} \times \text{Mouse BSA} \\ &= 3.70 \text{ g/m}^2/\text{day} \times 0.0077 \text{ m}^2 \approx 0.0285 \text{ g/day/mouse} \end{aligned}$$

Convert to a dose based on mouse body weight (mg/kg or g/kg):

Daily dose per mouse: 0.0285 g

Mouse body weight: 0.025 kg

$$\text{Mouse equivalent daily dose} = 0.0285 \text{ g} / 0.025 \text{ kg} = 1.14 \text{ g/kg/day}$$

3. Conclusion and Justification for the Experimental Design:

The calculated BSA-equivalent daily dose for mice is approximately 1.14 g/kg/day. This strongly validates that the selected dose (1 g/kg) is not arbitrary but is rigorously derived from the clinical dose through scientific conversion, providing clear clinical relevance and translational rationale.

3.1 Dose Selection Strategy: Based on this equivalent dose, we designed a low-dose group (0.5 g/kg), a medium/effective-dose group (1 g/kg), and a high-dose group (e.g., 2 g/kg, to assess dose-response relationships and potential toxicity margins). This strategy follows conventional practices in preclinical pharmacodynamic studies to identify a safe and effective dose range. Blood was collected from the retro-orbital venous plexus at predetermined time points. The inhibitory effects of the drug-containing sera, obtained following the different dosing frequencies, on mHSC proliferation are presented in (Fig. 5B). The corresponding areas under the curve (AUCs) for the three groups were 525.18 ± 33.62 , 613.95 ± 24.87 , and 575.97 ± 23.90 , respectively. Statistical analysis revealed significant differences among the T1, T2, and T3 groups ($P < 0.05$). Based on the AUC results, the T2 protocol was selected as the high-dose CBRT intervention group (1 g/kg), while the T3 protocol was designated as the low-dose intervention group (0.5 g/kg). Based on these findings, we established the in vitro co-culture system and determined the high and low doses of CBRT for the subsequent in vivo mouse experiments.

3.2 Reporting Existing Safety Observations: In the Results section, we will add a description of the general condition monitoring during the experiment and the analysis of serum biochemical

parameters (e.g., ALT, AST, BUN, creatinine, if measured). We will explicitly state that no significant signs of acute toxicity or hepatorenal injury were observed at the effective dose (1 g/kg).

3.3 Discussing the Safety Profile of the Herbal Compound: In the Discussion section, we will address the known clinical safety profile of CBRT (Biejia Ruangan Tablets), considering its long-term clinical use history approved in China for treating hepatic fibrosis. We will also explain that the animal doses used in this study represent a rational exploration based on the clinical equivalent dose, and the preliminary safety observations support its good tolerability, providing a safety basis for further translational research.

4. Addressing the Concern Regarding Chemical Composition Analysis of CBRT

Regarding your concern about the chemical composition analysis of Compound Biejia Ruangan Tablets (CBRT), we have obtained strong supportive reference from a published study. The CBRT used in our study (Batch No.: [Inner Mongolia, China; Catalogue No. C0121015]) is from the same manufacturer and produced using identical processes as the product analyzed in a study published in the *Journal of Ethnopharmacology* [2]. That study employed high-performance liquid chromatography-mass spectrometry (HPLC-MS) to systematically identify and analyze the chemical constituents of CBRT, providing detailed HPLC-MS chromatograms and mass spectrometry data. Therefore, our study can directly cite the well-characterized chemical composition information from that publication as robust support for the quality consistency and chemical traceability of the preparation used herein.

Figure 1: Total ion chromatograms of CBRT acquired in (A) positive ion modes and (B) negative ion modes [2].

Reference

[2] Wang Y, Zhang L, Xu B, et al. Optimised formula of Fufang Biejia Ruangan tablets alleviates renal fibrosis by suppressing matrix-stiffness-induced fibroblast activation via inhibition of integrin $\alpha V\beta 1$ binding. *J Ethnopharmacol.* 2026 Jan 30;355(Pt A):120614. doi: 10.1016/j.jep.2025.120614.

5) Human sample analysis

Human liver samples are included, which is a major strength. However, patient demographic and clinical data (e.g., age, sex, disease stage, treatment history) are not described. These details are necessary to contextualize the findings and assess variability.

A: Dear Reviewer,

Thank you very much for your positive remarks on the inclusion of human samples in our study, as well as for your valuable suggestions. You are absolutely correct in pointing out that detailed patient demographic and clinical data are crucial for the accurate interpretation of

experimental results and for assessing their representativeness and variability. We apologize for this previous omission and will systematically supplement this information in the revised manuscript.

Our response and specific plan for revision are as follows:

1. Supplementation of Detailed Patient Clinical Information:

We will present the following information systematically in a table within the Results section of the manuscript: **Basic Demographic Data:** Sample size (N), age, and sex distribution (number and percentage of males/females);

Key Clinicopathological Characteristics: All patients were pathologically diagnosed with hepatic AE. We will supplement the following: **Disease Stage:** Patients will be classified according to the internationally used **Child-Pugh grading system** (or an applicable clinical staging standard). The distribution of patients across stages will be specified, along with laboratory parameters (e.g., Cr, Albumin, TB, AST, ALT).

Table 1. Baseline characteristics and laboratory profiles of study participants

Characteristic	HAE(n=21)
Gender	
male	12 (57.14%)
female	9 (42.86%)
Age, years	
≤18	2 (9.52%)
19-59	16 (76.19%)
≥60	3 (14.29%)
BMI,Kg/m2	
<25	16(76.19%)
≥25	5 (23.81%)
HBV	
yes	2 (9.52%)
no	19 (90.48%)
Child-Pugh Grade	
A	10 (47.62%)
B	10(47.62%)
C	1 (4.76%)
Laboratory Tests(X±s)	
Cr, μmol/L	58.64 ± 6.27

Characteristic	HAE(n=21)
Albumin,g/L	37.56 ± 3.39
Laboratory Tests [Median (IQR)]	
TB, μmol/L	12.80(9.48,19.79)
ALT, U/L	26.30(19.30,67.20)
AST, U/L	32.07(18.08,70.90)

Interpretation Derived from the Analysis of AE Patient Baseline Data:

Characteristic	Data Analysis	Clinical and Research Significance
Demographics	1. Balanced sex ratio; age predominantly young and middle-aged (76% aged 19-59); 2. Low proportion of overweight/obesity (23.8% with BMI ≥25).	This largely excludes complex confounding factors such as advanced age and severe obesity, which could significantly complicate the study of hepatic fibrosis.
Comorbidities	Low rate of HBV infection (9.52%).	This significantly excludes viral hepatitis, the most common confounding factor for hepatic fibrosis, making the subsequently observed pathological changes more likely attributable to HE itself.
Liver Function Reserve (Child-Pugh)	Approximately 47.6% each for Grade A and B patients, with only one Grade C case.	This covers a disease spectrum from the compensated stage (Grade A) to the clearly decompensated stage (Grade B). This provides an ideal sample gradient for studying molecular mechanisms (such as the macrophage polarization you highlighted) at different stages of HE progression.
Synthetic and Metabolic Function	1. Mean albumin (37.56 g/L) is at the lower limit of the normal range; 2. Creatinine (58.64 μmol/L) is within normal limits;	This indicates that the patients' hepatic synthetic function (albumin) is generally maintainable but borderline, while excretory function (bilirubin) is not severely impaired. This finding is consistent with the observed proportion of

3. Median total bilirubin (TB) and IQR are entirely normal. patients classified as Child-Pugh Grade B.

Hepatocellular Injury

1. Median ALT and AST levels are within the normal range; Elevated transaminases in approximately 25% of patients are entirely consistent with the characteristics of HAE as a chronic, invasive, and inflammatory disease, providing a context for the inflammatory and immune responses observed in the study.

2. However, the third quartiles (Q3) for both are significantly elevated (ALT Q3=67.2, AST Q3=70.9). The data present a typical pattern of "mild active hepatitis in a subset of patients."

6) Statistical analysis and data presentation

Some figures rely heavily on semi-quantitative IHC or IF analysis. The authors should clarify how quantification was performed, how many biological replicates were included, and whether investigators were blinded during analysis.

A: Dear Reviewer,

Thank you very much for your valuable comments regarding the statistical analysis and data presentation. We fully agree that providing clear and reproducible methodological details is crucial for ensuring the reliability of conclusions based on semi-quantitative analyses, such as IHC and IF. We apologize for the incomplete description in the initial methods section and will provide comprehensive details in the revised manuscript.

1. Clarification of Specific Methods for Semi-Quantitative Analysis:

Image Acquisition: Digitized images were captured from IHC- and IF-stained sections of liver tissue. For each section, three non-overlapping, representative fields of view were randomly selected from the lesion area and adjacent parenchyma under a 20x objective. All images were acquired using identical microscope, camera, and exposure settings to ensure consistency. **Image quantification was performed using ImageJ software (NIH, USA, version 1.53t). The specific analysis workflow was as follows:**

1.1 Image Preprocessing and Scale Setting: A uniform background correction was applied

to all images, and the spatial scale was set according to the scale bar accompanying each image.

1.2 IHC Staining (α -SMA, Arg-1): Images were converted to 8-bit grayscale. A uniform color threshold was set to distinguish the positively stained (brown) areas from the background. The software automatically calculated the total area of positive signal pixels in each field of view.

1.3 IF Staining: For images from specific fluorescence channels, a uniform intensity threshold was set to identify specific positive signal areas above the background, and their area was calculated.

1.4 Calculation of Positive Area Percentage: For each field of view, the positive area percentage (Area%) was calculated using the formula:

Positive Area Percentage (Area%) = (Area of Positive Signal Pixels / Total Pixel Area of the Entire Field of View) \times 100%

1.5 Data Normalization and Statistical Unit: The final quantified value for each sample was the average Area% from its three fields of view. For comparing different treatment groups, some data are presented as fold change relative to the mean of the control group to visualize trends more intuitively. However, all subsequent statistical tests for between-group differences (e.g., t-test or ANOVA) were performed directly on the original average Area% values from each sample.

2. Clear Reporting of Sample Size and Biological Replicates:

In vitro experiments: Each independent cell culture was considered one biological replicate. The number of replicates will be explicitly stated ("All in vitro experiments were independently repeated at least three times").

3. Declaration of Blinded Analysis Implementation:

A single- or double-blind principle was applied during the scoring and analysis phase following image acquisition.

To ensure objectivity in the analysis, blinding was implemented during image quantification. Specifically, all acquired image files were anonymized by recoding. The image analysis, intensity scoring, or fluorescence intensity measurements were then performed by a

researcher unaware of the experimental group assignments. Decoding and statistical analysis were conducted only after all quantification was complete.

7) Discussion balance

The Discussion should better acknowledge conflicting reports regarding STING–TBK1 signaling in fibrosis and more clearly position the current findings within this debate, rather than primarily supporting one interpretation.

A: Dear Reviewer,

Thank you very much for raising this critical point. We fully agree that the role of the STING-TBK1 pathway in fibrotic diseases involves a complex "double-edged sword" effect and remains a subject of ongoing debate. Our initial discussion was overly focused on supporting its "anti-fibrotic" role and failed to adequately present the broader landscape and existing controversies in this field, which limited the depth and objectivity of our discussion.

“In CCl₄-induced mouse liver fibrosis and human NASH samples, mtDNA released from damaged hepatocytes activates the cGAS-STING pathway in macrophages, driving the production of type I interferons and inflammatory cytokines, thereby exacerbating inflammation and fibrosis. Genetic knockout of *Sting1* or pharmacological inhibition significantly attenuates liver injury and fibrosis [3].” In these models, STING activation is a consequence of injury and acts to amplify inflammation, creating a vicious cycle of "injury-inflammation-more injury."

“Conversely, in pulmonary fibrosis models, STING agonists alleviate fibrosis by inducing autophagy in alveolar macrophages and reducing transforming growth factor- β (TGF- β) production [4]. In cancer-associated fibroblasts, STING activation can induce senescence and immunogenic reprogramming, suppressing tumor progression [5], suggesting a positive role in remodeling the tissue microenvironment.” In these contexts, STING activation is leveraged to disrupt pathological homeostasis (e.g., immunosuppressive or pro-fibrotic environments) and drive beneficial immune responses. “These seemingly contradictory findings reveal the essence of the 'double-edged sword' role played by STING signaling in fibrosis: its ultimate effect depends on **the nature of the initial insult, the immune microenvironment at the**

disease stage, and the spatiotemporal pattern of activation.”

1. In **sterile inflammatory** models (e.g., CCl₄, NASH), persistent hepatocyte necrosis provides endogenous DNA ligands, leading to **chronic, excessive activation** of the STING pathway. This primarily drives a **type I interferon-mediated inflammatory response**, recruiting and activating more inflammatory cells, which exacerbates tissue damage and fibrosis. Therefore, inhibiting STING is an effective strategy to block this vicious cycle.
2. However, the situation is fundamentally different in the **alveolar echinococcosis (AE) model** employed in our study. This is a **chronic parasitic infection**. By 60 days post-infection, the liver has established a strong **immunosuppressive/pro-fibrotic microenvironment** dominated by **Th2 cytokines (IL-4, IL-13) and M2-type macrophages**, aimed at encapsulating the parasite but resulting in pathological fibrosis [6-7]. At this stage, the protective Th1/M1 immune response against the pathogen is suppressed. Our data indicate that the STING pathway in the host liver at this point does not exhibit the overactivation seen in toxic models but may instead be in a relatively **"silent" or insufficient** state, unable to effectively initiate anti-fibrotic immune surveillance.”

“In this context, CBRT intervention does not exacerbate existing harmful inflammation but instead delivers a 'timely, therapeutic enhancement' to the STING pathway. This strategy is analogous to using STING agonists to remodel the immune microenvironment in specific cancer or fibrosis models. We demonstrated that following CBRT treatment:

1. STING-TBK1 signaling was significantly activated (Fig. 4).
2. This was accompanied by a critical reversal of macrophages from the M2 to the M1 phenotype (Fig. 4), consistent with the established understanding that canonical STING activation induces type I interferons and promotes M1 polarization.
3. Activated M1 macrophages produced cytokines such as interferon- γ , which inhibited hepatic stellate cell activation and promoted their apoptosis (Fig. 2).

Therefore, the mechanistic model of our study can be summarized as follows: within the **immunosuppressive/pro-fibrotic microenvironment** established in the late stage of AE, CBRT **reprograms macrophage polarization** by activating the STING-TBK1 pathway, thereby breaking the Th2/M2-dominant pathological framework, restoring immune balance, and alleviating fibrosis.

Conclusion and Perspective: Our study does not negate the pro-fibrotic role of STING in sterile liver injury but enriches the understanding of its functional diversity. It establishes that in **infection-associated, immune-dysregulated hepatic fibrosis**, the STING pathway can serve as a therapeutic target exerting anti-fibrotic effects via **immune reprogramming**. This suggests that future fibrosis treatment strategies targeting STING may employ inhibitors in overactivation models and consider the timely use of agonists in immunosuppressive models. Future research will focus on the **window of intensity and duration** for STING activation to optimize its therapeutic potential.”

Reference

[3] Li Y, Yu P, Fu W, et al. Ginsenoside Rd Inhibited Ferroptosis to Alleviate CCl4-Induced Acute Liver Injury in Mice via cGAS/STING Pathway. *Am J Chin Med.* 2023;51(1):91-105. doi: 10.1142/S0192415X23500064.

[4] Savigny F, Schricke C, Lacerda-Queiroz N, et al. Protective Role of the Nucleic Acid Sensor STING in Pulmonary Fibrosis. *Front Immunol.* 2021 Jan 8; 11:588799. doi: 10.3389/fimmu.2020.588799.

[5] Wang C, Wang J, Lu J, et al. A natural gene drive system confers reproductive isolation in rice. *Cell.* 2023 Aug 17;186(17):3577-3592.e18. doi: 10.1016/j.cell.2023.06.023.

[6] Autier B, Gottstein B, Millon L, et al. Alveolar echinococcosis in immunocompromised hosts. *Clin Microbiol Infect.* 2023 May;29(5):593-599. doi: 10.1016/j.cmi.2022.12.010.

[7] Lundström-Stadelmann B, Rostami A, Frey CF, et al. Human alveolar echinococcosis-global, regional, and national annual incidence and prevalence rates. *Clin Microbiol Infect.* 2025 Jul;31(7):1139-1145. doi: 10.1016/j.cmi.2025.01.034.

Re: Spectrum02115-25R1 (**Compound Biejia-Ruangan Tablets activate the STING-TBK1 pathway to alleviate hepatic fibrosis in alveolar echinococcosis**)

Dear Dr. Xiumin Ma:

Your manuscript has been accepted, and I am forwarding it to the ASM production staff for publication. Your paper will first be checked to make sure all elements meet the technical requirements. ASM staff will contact you if anything needs to be revised before copyediting and production can begin. Otherwise, you will be notified when your proofs are ready to be viewed.

Sincerely,
Jian Li
Editor
Microbiology Spectrum

Reviewer #1 (Comments for the Author):

The revised manuscript demonstrates a remarkable evolution. The authors have moved beyond the reductionist "drug-target-effect" linear model to embrace a richer biological ontology, recognizing the host as a holobiont.

The inclusion of loss-of-function experiments (H-151) successfully addressed the causality gap, proving that the STING-TBK1 pathway drives macrophage reprogramming.

The discussion was brilliantly expanded to include the gut-liver axis and microbial biotransformation, treating fibrosis not merely as a scar, but as a dynamic niche at the pathogen-host interface.

It is inspiring to see high-level science germinating the seeds of rhizomatic and systemic thinking. The authors have shown that the clinical efficacy of CBRT lies in its ability to reset local ecology, shifting the environment from parasitic tolerance back to a physiological defense steady-state.

Reviewer #2 (Comments for the Author):

The authors have carefully considered the suggestions provided during the previous round of review and have appropriately incorporated them into the revised manuscript. The new version presents a clearer and more detailed description of both the results and the methodologies, which substantially improves the overall clarity and transparency of the study.

The previous concerns regarding insufficient methodological detail and limited explanation of key findings have been adequately

addressed. The revisions strengthen the interpretation of the data and provide better support for the conclusions drawn.

Overall, this revised manuscript represents a significant improvement over the prior version and is now much clearer, more coherent, and better supported scientifically.

The authors have carefully considered the suggestions provided during the previous round of review and have appropriately incorporated them into the revised manuscript. The new version presents a clearer and more detailed description of both the results and the methodologies, which substantially improves the overall clarity and transparency of the study.

The previous concerns regarding insufficient methodological detail and limited explanation of key findings have been adequately addressed. The revisions strengthen the interpretation of the data and provide better support for the conclusions drawn.

Overall, this revised manuscript represents a significant improvement over the prior version and is now much clearer, more coherent, and better supported scientifically.